# Lockdown: Backdoor Defense for Federated Learning with Isolated Subspace Training

**Tiansheng Huang, Sihao Hu, Ka-Ho Chow, Fatih Ilhan, Selim Furkan Tekin, Ling Liu**
School of Computer Science
Georgia Institute of Technology, Atlanta, USA
{thuang374, shu335, kchow35, filhan3, stekin6}@gatech.edu, ling.liu@cc.gatech.edu

## Abstract

Federated learning (FL) is vulnerable to backdoor attacks due to its distributed computing nature. Existing defense solution usually requires larger amount of computation in either the training or testing phase, which limits their practicality in the resource-constrain scenarios. A more practical defense, i.e., neural network (NN) pruning based defense has been proposed in centralized backdoor setting. However, our empirical study shows that traditional pruning-based solution suffers *poison-coupling* effect in FL, which significantly degrades the defense performance. This paper presents Lockdown, an isolated subspace training method to mitigate the poison-coupling effect. Lockdown follows three key procedures. First, it modifies the training protocol by isolating the training subspaces for different clients. Second, it utilizes randomness in initializing isolated subspacess, and performs subspace pruning and subspace recovery to segregate the subspaces between malicious and benign clients. Third, it introduces quorum consensus to cure the global model by purging malicious/dummy parameters. Empirical results show that Lockdown achieves *superior* and *consistent* defense performance compared to existing representative approaches against backdoor attacks. Another value-added property of Lockdown is the communication-efficiency and model complexity reduction, which are both critical for resource-constrain FL scenario. Our code is available at `https://github.com/git-disl/Lockdown`.

## 1 Introduction

Federated Learning (FL) (McMahan et al., 2016) is a privacy-preserving machine learning paradigm that allows the training surrogates (clients) to collectively train a global model with data in local devices. However, because the training data and potentially the training process on the clients lacks supervision, it is possible that attackers can launch data poisoning attack on the global model (Tolpegin et al., 2020), so as to manipulate the prediction of the model.

Backdoor attack is one of the data poisoning attacks, which is stealthy and disruptive to the normal behavior of the model. Specifically, the prediction of the model can be manipulated such that it consistently predicts one (or some) chosen

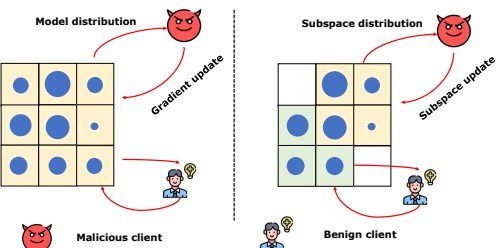

Figure 1: Illustration of isolated subspace training. The blue circles and the shaded area respectively represent the parameters of the model and the training subspace. Left: vanilla FL where malicious clients can poison all the parameters. Right: isolated subspace training where malicious clients can only poison a subspace of them.

label whenever it is given samples with a backdoor trigger. Backdoor attack poses serious threats to

37th Conference on Neural Information Processing Systems (NeurIPS 2023).

many security-critical applications, e.g., biometric authentication and autonomous driving (Chow & Liu, 2021), which vitiates the large-scale deployment of FL. Therefore, an effective defense that can mitigate such a risk is in an urgent need.

The main stream of existing research to defend backdoor attack in FL can be mainly classified into three genres, i) certified robustness (Xie et al., 2021), (Alfarra et al., 2022), ii) adversarial training (Zizzo et al., 2020; Shah et al., 2021), and iii) robust aggregation (Chow et al., 2023; Zhang et al., 2022a). Though existing defenses can mitigate attack, they are still far from their maturity. In particular, we highlight the *in-efficiency* of state-of-the-art defense solutions.

*Existing defenses usually require extra computation in either the training or inference phases.* For example, randomized smoothing, the key component for certified robustness defense, requires multiple forward pass of data in the inference phase for producing one effective prediction. Analogously, adversarial training requires to generate adversarial samples in the inner loop of training, and therefore demands extra data pass. This issue is particularly pronounced in the FL scenario, considering that training and the inference are conducted on edge devices with limited capacity (Wang et al., 2019b).

In this paper, we try to answer the following question:

> *Is there a solution that not only provides SOTA defense performance, but also requires equal or even less computation in both training and deployment phases?*

To this end, we first apply the pruning-based defense solution in FL backdoor setting, but we observe a poison-coupling effect, which significantly degrades the defense performance. Driven by this observation, we propose Lockdown, a solution that utilizes isolated subspace training to de-couple the poison parameters, which are then pruned using the consensus from clients. Our experiments demonstrate that: i) Lockdown reduces the Attack Success Ratio (ASR) by up-to 90% compared to FedAvg without defense. ii) Lockdown consistently performs better than SOTA defense solutions in various attack settings. Specifically, Lockdown acquires 83% of ASR reduction with only 2.7% drop of benign accuracy when the data distribution is Non-IID (the other two defense baselines RLR and Krum respectively acquire 14.2% and 86.1% ASR reduction while sacrificing 16.2% and 43.6% benign accuracy in the same setting). iii) Lockdown reduces both downlink and uplink communication by at least 0.75x, and the number of parameters used in the model training and inference phases are also accordingly reduced by at least 0.75x thanks to the removal of malicious/dummy parameters.

To the end, we summarize our contribution as follows:

- We study the data-free pruning defense in FL setting, and find that the poison parameters tend to be statistically coupled with benign parameters, which we refer to as "poison-coupling" effect in FL backdoor. To the best of our knowledge, this finding is not available in the literature.

- We propose Lockdown, a backdoor defense that utilizes the idea of isolated subspace training to decouple the poison parameters. Notably, Lockdown i) enjoys communication reduction between server and clients, and ii) lowers the model's training/inference complexity.

- We conduct evaluations to show the efficacy of Lockdown. Results show that Lockdown *consistently* outperforms existing defense baselines under different attack settings (attack method, attacker number and poison ratio) and different data distributions (IID and Non-IID).

- Ablation study and hyper-parameter sensitivity analysis are conducted to verify the individual functionality of each component of Lockdown.

## 2 Related Work

**Federated Learning**. Federated learning (McMahan et al., 2016) is a privacy-preserving distributed training paradigm that allows clients to collectively train a global model from distributed training data. Recent works on FL mostly lie in optimization (Karimireddy et al., 2020; Li et al., 2018; Sun et al., 2023a,c; Li et al., 2023b; Sun et al., 2023b), which study how to mitigate the Non-IID issue (Zhao et al., 2018) and system efficiency (Huang et al., 2020, 2022a; Li et al., 2021b,a).

**Federated backdoor attack and defense**. Classical backdoor attack in FL is empirically proven to be effective in (Bagdasaryan et al., 2020), and subsequently, new types of attack are developed, e.g., edge-case backdoor (Wang et al., 2020), stealthy model poisoning (Bhagoji et al., 2019), subnet replacement attack (Qi et al., 2022), and Distributed Backdoor Attack (Xie et al., 2019).

Backdoor defense are developed to counter the potential threat posed by the backdoor attacks (representative methods are neural cleanse (Wang et al., 2019a), meta analysis (Xu et al., 2021), etc). Under the FL setting, backdoor defenses can be classified into three main genres. The first genre is certified robustness. Certified robustness relies on randomized smoothing(Cohen et al., 2019), an approach with theoretical certification of the model robustness. Subsequent techniques, e.g., weight smoothing (Xie et al., 2021) and group ensemble (Cao et al., 2022) are studied for the defense of federated backdoor. The second genre is adversarial training (Zizzo et al., 2020; Shah et al., 2021), which generates adversarial samples to improve the model's robustness. However, both certified robustness and adversarial training require more computation in either the training or deployment stage. The third genre is robust aggregation, which studies how to identify and preclude the malicious update in the aggregation stage (Chow et al., 2023; Zhang et al., 2022a; Guo et al., 2021; Panda et al., 2022). However, the classification benign accuracy of this category of defense often perform worse when Non-IID extent increases.

**Sparse training**. Sparse training is originally designed to reduce the model complexity for both the training and deployment stage. SET (Mocanu et al., 2018) first proposes the idea of dynamic subspace searching with alternative pruning and recovery process, and are empirically studied with the concept of in-time over-parameterization (Liu et al., 2021b,a, 2022). (Evci et al., 2020) further introduces gradient information to guide the mask searching process. This model compression technique has recently been extended to FL (Bibikar et al., 2022; Huang et al., 2022b; Dai et al., 2022).

We utilize sparse training technique to develop a new genre of backdoor defense solution. To the best of our knowledge, this is the first attempt that connects sparse training with backdoor defense in FL.

## 3 Threat Models

We consider three threat models, named *weak*, *medium* and *strong*. All the models allow multiple attackers to co-exist in the system. We use $N$ to denote the number of attackers out of $M$ total clients.

Table 1: Permission of threat models.

| Permission\Threat model | weak | medium | strong |
|---|:---:|:---:|:---:|
| Data manipulation | ✓ | ✓ | ✓ |
| Algorithm manipulation | ✗ | ✓ | ✓ |
| Global information Access | ✗ | ✗ | ✓ |
| Aggregation manipulation | ✗ | ✗ | ✗ |

**Permission.** Different threats models are given different control permissions. Specifically, *Weak* model allows the attackers to arbitrarily manipulate its local data, but cannot control its local training process, while attackers in *medium* and *strong* model are allowed to do so. Only *strong* model allows the attackers to obtain other benign client's information, e.g., update from them. For control permission in the weak threat model, technique like trusted execution environment (Mo et al., 2021) can be applied to control each client to run the designated program, and thereby disabling algorithm manipulation. Existing literature usually adopt medium threat model, in which attackers can do whatever they like in their local devices, but has not extra information from server or from other benign clients. The permission of the threat models are summarized in Table 1.

**Malicious objective.** Denote the set of benign clients as $\mathcal{M}$ and the set of malicious clients as $\mathcal{N}$. Formally, we characterize the objective of malicious clients as follows:

$$\min_{\boldsymbol{w}} \frac{1}{M} \left( \sum_{i \in \mathcal{N}} \tilde{f}_i(\boldsymbol{w}) + \sum_{i \in \mathcal{M}/\mathcal{N}} f_i(\boldsymbol{w}) \right) \tag{1}$$

where $\tilde{f}_i(\boldsymbol{w}) \triangleq \frac{1}{|\tilde{\mathcal{D}}_i|} \sum_{(\boldsymbol{x}_j, y_j) \in \tilde{\mathcal{D}}_i} CE(\boldsymbol{w}; \boldsymbol{x}_j, y_j))$ is the malicious objective of an attacker (to minimize the average cross-entropy of the backdoor dataset), $f_i(\boldsymbol{w}) \triangleq \frac{1}{|\mathcal{D}_i|} \sum_{(\boldsymbol{x}_j, y_j) \in \mathcal{D}_i} CE(\boldsymbol{w}; \boldsymbol{x}_j, y_j))$ is the benign objective (to minimize the average CE loss for the dataset from benign clients) , and $\tilde{\mathcal{D}}_i$ and $\mathcal{D}_i$ are respectively the backdoor dataset and pristine dataset.

**Attack methods.** Different attack methods are eligible in different threat models. For weak threat model, only data-level backdoor, e.g., BadNet (Gu et al., 2017), DBA (Xie et al., 2019), and Sinusoidal

(Barni et al., 2019) can be applied, since the attacker has no control to the local training, but only has access to the dataset. For medium threat model, the attacker can perform a wider range of attack methods, e.g., Scaling (Bagdasaryan et al., 2020), FixMask, etc, which requires the modification of the local training process. For the strong threat model, we test Neurontoxin (Zhang et al., 2022b), and Omniscient, which requires extra server-side information to aid the attack.

**Defense goal.** While solving the benign objective (see FL general objective in (McMahan et al., 2016)), we expect the global model $w$ to be able to resist backdoor attack. In other words, the global model should minimize the benign objective while maximizing the malicious objective.

## 4 Case Study on Pruning-based Defense

Motivated by our goal to propose a computation-friendly defense, we extend the pruning-based defense originally proposed for centralized ML to federated learning setting. We study CLP defense proposed in (Zheng et al., 2022) towards two poisoned models trained with centralized SGD (centralized backdoor) and the global model produced by FedAvg (federated backdoor) [1]. Surprisingly, we observe that the pruning defense exhibits substantially different performance on the two models.

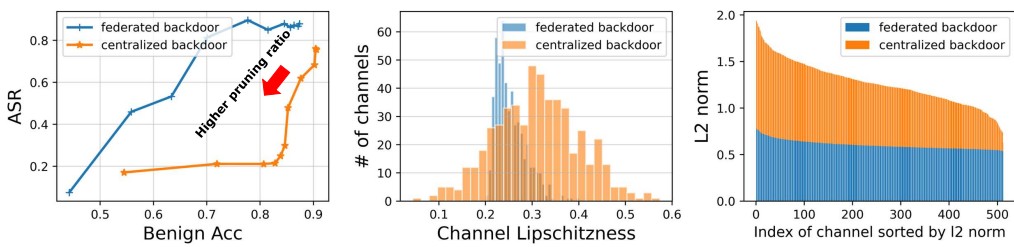

Figure 2: Properties of two models trained with centralized backdoor and federated backdoor. Left: ASR and benign accuracy with CLP defense in (Zheng et al., 2022). Middle: Channel lipschitzness of last convolutional layer of two models. Right: L2 norm of last convolutional layer of two models. The experiment is done on a ResNet-9 model on CIFAR10 dataset and under BadNet attack.

**Observations.** In the left of Figure 2, we observe that the pruning algorithm cannot efficiently eliminate the backdoor parameters for federated backdoor. The ASR can only drop to an acceptable range (say 30%) when leveraging large pruning ratio, and accordingly, this also leads to a large drop of benign accuracy (say 35% drop). In the middle, we plot the channel lipschitzness (CL) of the two models, and find that the CL values for a federated backdoor couple in a more compact range, which makes it hard for the pruning method to find out the malicious channels to prune. This explains why CLP fails to work for federated backdoor. In the right, we plot the L2 norm of weights of the last convolutional layer, the same coupling effect for federated backdoor can be observed – the L2 norm of different channels are all within a small range.

**Poison-coupling effect.** In summary, our main takeaway is that the federated backdoor model is hard to be cured by classical pruning method, because the backdoor parameters does not exhibit substantial statistical difference with the benign ones, i.e., it tends to be coupled with the benign parameters. We refer to this phenomenon for federated backdoor as *poison-coupling* effect. Due to the existence of poison-coupling effect, it is hard, if not impossible, to accurately identify the poisoned parameters after the model has been fully trained by federated learning process, and thereby reducing the efficiency of the pure parameters pruning defense.

## 5 Methodology

To mitigate the *poison-coupling* effect, we propose lockdown with isolated subspace training. Intuitively, we don't allow the malicious clients to get access to all the parameters, but *only a subspace of them* (See Figure 1), in order to prevent them from statistically coupling the poisoned parameters with the benign ones that are supposed to do normal function.

---

[1]Checkpoints of the centralized/federated backdoor models are available in `https://www.dropbox.com/scl/fo/lhi0objyklgnyh299j5wz/h?rlkey=wrsa5msbl2m3kzuk2f8rcc8uw&dl=0`

The high-level idea of Lockdown is as follows. i) we employ *isolated subspace training* to restrain the training of each client into its own subspace, such that the backdoor data cannot access and couple the poisoning function in all the parameters. ii) we employ *dynamic subspace searching* for the clients to search for local subspaces using their local datasets, which involve only the parameters that they deem important. iii) After the above local procedures, the server will *aggregate* the gradient updates into the corresponding subspaces of global model, and continues the next round of training. iv) Repeating $T$ rounds of training, the server can identify the malicious parameters by *consensus fusion*. Under the intuition that the malicious/dummy parameters (at specific coordinates) should have less chance to

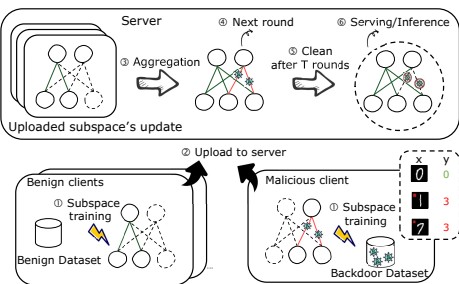

Figure 3: Overview of Lockdown. First, clients receive subspace model from server, and apply isolated subspace training. Second, clients upload their subspace update. Third, server aggregates the updates into global model. After training, consensus fusion is applied to remove poison params.

be involved in benign clients' subspaces (because they are not important parameters for the benign datasets). Considering the benign clients constitute the majority in the system, consensus fusion identify those parameters that have the least appearance among all the clients subspace as the malicious or dummy parameters, which are then pruned to mitigate backdoor behavior. As follows, we detail each step of Lockdown.

**Subspace initialization.** We enforce an overall sparsity $s$ to the initial subspace of each client (denoted by a binary variable $\boldsymbol{m}_{i,0}$). For maintaining practical performance of sparsity, we follow (Evci et al., 2020) to use ERK for random subspace initialization, in which different layers in the corresponding subspace are designated different sparsity (See Appendix A.1 for details).

For each client, we enforce them to have the same initial subspace, which can be achieved via enforcing the same random seed to them for subspace generation. However, we do find that heterogeneous mask initialization can increase protection in some cases (See Section 6.2).

**Isolated subspace training.** In this stage, each client performs multiple local steps to train parameters within its isolated subspace (as enforced by its mask). Specifically, for local step $k = 0, \ldots, K - 1$, clients do this update:

$$\boldsymbol{w}_{i,t,k+1} = \boldsymbol{w}_{i,t,k} - \eta \boldsymbol{m}_{i,t} \odot \nabla f_i(\boldsymbol{w}_{i,t,k}; \xi) \tag{2}$$

where $\xi$ is a piece of random sample within the local dataset, $\eta$ is the learning rate, and $\odot$ denotes Hadamard product. As shown, the binary mask $\boldsymbol{m}_{i,t}$ is applied to the gradient in every local step, which isolates the training into $i$-th client's own subspace. This process prevents the backdoor data from contaminating all the parameters within the parameter space.

**Subspace searching.** We conduct subspace searching for clients to search for their subspaces, i.e., the parameters that are important per the data they have. The searching process is decomposed to two phases, as follows:

---

**Algorithm 1** Lockdown defense

**input** Training iteration $T$; Local steps $K$; Learning rate $\eta$; Pruning/recovery rate $\alpha_t$ decayed by cosine annealing (Specially, $\alpha_{-1} = 0$, i.e., no recovery for first round); Random seed $seed$; Initial model $\boldsymbol{w}_0$.
**output** Clean model for deployment $\tilde{\boldsymbol{w}}_T$
 1: **main** Server's Main Loop
 2:   $\boldsymbol{m}_{i,0} = $ **SubspaceInit**($seed$) for $i \in \mathcal{M}$
 3:   **for** $t = 0, 1, \ldots, T - 1$ **do**
 4:     **for** $i \in \mathcal{M}$ **do**
 5:       Send $\boldsymbol{w}_{i,t,0} = \boldsymbol{m}_{i,t} \odot \boldsymbol{w}_t$ to client $i$
 6:       Call Client $i$'s main loop for training
 7:       Received $\boldsymbol{w}_{i,t,K}$ and $\boldsymbol{m}_{i,t+1}$
 8:     **end for**
 9:     $\boldsymbol{w}_{t+1} = \boldsymbol{w}_t - \frac{1}{M} \sum_{i=1}^{M} \boldsymbol{m}_{i,t} \odot (\boldsymbol{w}_t - \boldsymbol{w}_{i,t,K})$
10:   **end for**
11:   $\tilde{\boldsymbol{w}}_T = \boldsymbol{w}_T \odot \mathcal{T}_\theta(\boldsymbol{m}_{1,T}, \ldots, \boldsymbol{m}_{M,T})$
12:   Deploy $\tilde{\boldsymbol{w}}_T$ for serving/inference.
13: **end main**
14: **main** Client's Main Loop
15:   Obtain local gradient $\nabla f_i(\boldsymbol{w}_{i,t,0})$
16:   $\boldsymbol{m}_{i,t+\frac{1}{2}} = \boldsymbol{m}_{i,t} + \text{ArgTopK}_{\alpha_{t-1}}(|\nabla f_i(\boldsymbol{w}_{i,t,0})|)$
17:   **for** $k = 0, 1, \ldots, K - 1$ **do**
18:     $\boldsymbol{w}_{i,t,k+1} = \boldsymbol{w}_{i,t,k} - \eta \boldsymbol{m}_{i,t+\frac{1}{2}} \odot \nabla f_i(\boldsymbol{w}_{i,t,k}; \xi)$
19:   **end for**
20:   $\boldsymbol{m}_{i,t+1} = \boldsymbol{m}_{i,t+\frac{1}{2}} - \text{ArgBottomK}_{\alpha_t}(|\boldsymbol{w}_{i,t,K}|)$
21:   Send $\boldsymbol{w}_{i,t,K}$ and $\boldsymbol{m}_{i,t+1}$ to server
22: **end main**

---

1. **Subspace pruning.** In this phase, we prune the unimportant parameters within the client's current subspace that do not function at all. After $K$ steps of local training, we can identify the unimportant parameters as those with the smallest magnitude within the subspace. To preclude them from future training, we prune those smallest $\alpha_t$ parameters within each layer, and accordingly update

the mask to $\boldsymbol{m}_{i,t+1}$. Formally, this process can be formulated as follows:

$$\boldsymbol{m}_{i,t+1} = \boldsymbol{m}_{i,t+\frac{1}{2}} - \text{ArgBottomK}_{\alpha_t}(|\boldsymbol{w}_{i,t,K}|) \tag{3}$$

where $\text{ArgBottomK}_{\alpha_t}(|\boldsymbol{w}|)$ return the $\alpha_t$ percentage of smallest coordinates of each layer absoluted weights and mask them to 1, indicating that they will be pruned. We use cosine annealing to decay $\alpha_t$ from the initial pruning rate $\alpha_0$. The decay process is postponed to the Appendix A.1.

2. **Subspace recovery.** After pruning, and before the start of next round local training, the client recovers the same amount of parameters (as indicated by $\alpha_{t-1}$) to its subspace for exploration. To identify which parameters should be recovered, we use each client's data to extract the gradient w.r.t the weights after pruned, i.e., to extract $\nabla f_i(\boldsymbol{w}_{i,t,0})$. Then we recover $\alpha_{t-1}$ percentage of parameters by identifying those with top-$\alpha_{t-1}$ percentage of gradient magnitude within each layer, and accordingly update the mask to $\boldsymbol{m}_{i,t+\frac{1}{2}}$. The intuition is that for an important parameter over the local data, its gradient magnitude should be larger than the unimportant ones, which should be included into the new subspace. Formally, the recovery process is formalized as follows:

$$\boldsymbol{m}_{i,t+\frac{1}{2}} = \boldsymbol{m}_{i,t} + \text{ArgTopK}_{\alpha_{t-1}}(|\nabla f_i(\boldsymbol{w}_{i,t,0})|) \tag{4}$$

where $\text{ArgTopK}_{\alpha_{t-1}}(\boldsymbol{w})$ returns the $\alpha_{t-1}$ percentage of largest coordinates of absoluted gradient and mask them to 1, indicating that they will be recovered. But we note here that we don't perform subspace recovery in the first round, since pruning is not done yet.

**Aggregation.** Once subspace training is done, the clients upload the gradient updates of the local subspace to the server for aggregation. To aggregate the knowledge into the global model, we average the gradient updates based on their coordinate-wise contributions, as follows:

$$\boldsymbol{w}_{t+1} = \boldsymbol{w}_t - \frac{1}{M} \sum_{i=1}^{M} \boldsymbol{m}_{i,t+1} \odot (\boldsymbol{w}_t - \boldsymbol{w}_{i,t,K}) \tag{5}$$

where $\boldsymbol{w}_{i,t,K}$ is the weight from the client's local subspace, and $\boldsymbol{m}_{i,t+1}$ is the subspace after pruned.

**Consensus fusion (CF).** Given that those malicious parameters served to recognize backdoor triggers will be deemed unimportant for benign clients, they should not be contained in the subspace of benign clients, which accounts for the majority. This observation inspires us to eliminate the malicious parameters using consensus fusion after $T$ rounds of global model training. Formally, the consensus fusion operator returns a vector that satisfies:

$$[\mathcal{T}_\theta(\boldsymbol{m}_{1,T}, \ldots, \boldsymbol{m}_{M,T})]_j = \begin{cases} 1 & \sum_{i=1}^{M} [\boldsymbol{m}_{i,T}]_j \geq \theta \\ 0 & \text{Otherwise} \end{cases} \tag{6}$$

where $\theta$ is the threshold for CF, and $[\cdot]_j$ indexes the $j$-th coordinate of a vector. By applying the global mask produced by CF into the global model, i.e., $\boldsymbol{w}_T \odot \mathcal{T}_\theta(\boldsymbol{m}_{1,T}, \ldots, \boldsymbol{m}_{M,T})$, those parameters that have appearances smaller than $\theta$ among all the subspaces are sparsified to 0. In this way, those poisoned parameters will mostly be eliminated, thereby reaching the goal of perturbing backdoor.

## 6 Experiment

### 6.1 Experiment Setup

**Datasets and models.** We experiment on FashionMnist, CIFAR10/CIFAR100 and TinyImagenet datasets. For CIFAR10/CIFAR100, we use a ResNet9(He et al., 2016) as backbone. For TinyImagenet, we use a modified ResNet9 via adding a pooling layer after the first convolutional layer to keep the same hidden size before output. For FashionMnist, we use LeNet (LeCun et al., 1998).

**Attack methods.** We classify the backdoor attack methods in FL into three genres, i.e., *data-level backdoor*, *algorithm-level backdoor* and *advanced backdoor*. Data-level backdoor only allows the malicious clients to modify the raw data, but they have no control of the algorithm. On contrary, algorithm-level backdoor allows clients to modify the training algorithm, in addition to the raw data. The advanced attack can access extra server-side information (see Appendix A.2). Indeed, the three genres of backdoor attack methods correspond to the weak, medium, strong threat models in Table 1

respectively. Among the data-level backdoor, we simulate three types of attacks methods, i.e., BadNet (Gu et al., 2017), DBA (Xie et al., 2019), and Sinusoidal (Barni et al., 2019). For algorithm-level backdoor, we simulate two types of attack, i.e., Scaling (Bagdasaryan et al., 2020), and FixMask. For advanced backdoor, we test Neurotoxin (Zhang et al., 2022b) and an adaptive attack Omniscience.

**Defense Baselines.** We use vanilla FedAvg (McMahan et al., 2016) (without defense) as a baseline, and compare Lockdown with four SOTA defenses RLR (Ozdayi et al., 2021), Krum (Blanchard et al., 2017), RFA (Pillutla et al., 2022) and Trimmed mean (Yin et al., 2018).

**Evaluation metrics**. Three metrics are used to evaluate the defense performance:

- **Benign Acc.** Benign accuracy measures the Top-1 accuracy performance of a model given the benign data inputs (without the presence of a trigger).
- **ASR.** Attack Success Ratio (ASR) measures the ratio of backdoor samples (with trigger) to be classified to the target label. The smaller this metric, the more resilient the model is.
- **Backdoor Acc.** Backdoor accuracy measures the Top-1 accuracy of the model given the backdoor inputs (their labels during testing are the original one before adding backdoor trigger). We add this metric to evaluate the overall performance, since high backdoor acc means: i) the classification of benign features is well-performing. ii) the perturbation of backdoor trigger is limited.

**Simulation setting.** We simulate $M = 40$ clients, and data is either evenly distributed to each client (IID setting) or is distributed with Dirichlet distribution (Non-IID setting) following (Hsu et al., 2019). The parameter for Dirichlet distribution is set to 0.5 for the Non-IID partition. To simulate the backdoor attack launched by the malicious clients, we follow (Ozdayi et al., 2021) to randomly choose $N$ clients as attackers whose $p$ (percentage) of data in their local datasets are poisoned. The default backdoor settings for our main experiment are $p = 50\%$ and $N = 4$. We summarize the default simulation setting in Table 2. All the experiments are done with a Nvidia A100 GPU.

**Hyper-parameters.** For Lockdown, we fix the overall sparsity to $s = 0.25$, the mask agreement threshold to $\theta = 20$, and the initial pruning rate to $\alpha_0 = 1e - 4$. The robust learning rate threshold for RLR is set to 8. The number of local epochs and batch size are fixed to 2 and 64, respectively. The learning rate and weight decay used in the local optimizer are fixed to 0.1 and $10^{-4}$. The number of comm rounds is fixed to 200. We summarize the default hyper-parameters in Table 3.

| | Table 2: Default Simulation Setting. | | | Table 3: Default hyper-parameter for Lockdown. | |
|---|---|---|---|---|---|
| Notation | Meaning | Default Value | Notation | Meaning | Default Value |
| $M$ | Total number of clients | 40 | $\alpha_0$ | Initial pruning rate | 1e-4 |
| $p$ | Poison ratio | 0.5 | $\theta$ | Agreement threshold | 20 |
| $N$ | Attacker number | 4 | $s$ | Overall sparsity | 0.25 |

## 6.2 Main Evaluation

In our main evaluation, we use CIFAR10 as the default dataset and BadNet as the default attack.

**Convergence.** The convergence result w.r.t communication rounds is plotted in Figure 4. Lockdown offers the strongest robustness under attack with Non-IID settings. Compared with IID, when the data distribution is Non-IID, Lockdown suffers a slight drop in benign accuracy (by approximately 3%) while Krum and RLR suffer more. The larger heterogeneity causes more benign parameters to be dropped in the consensus fusion process, resulting in a drop in benign accuracy.

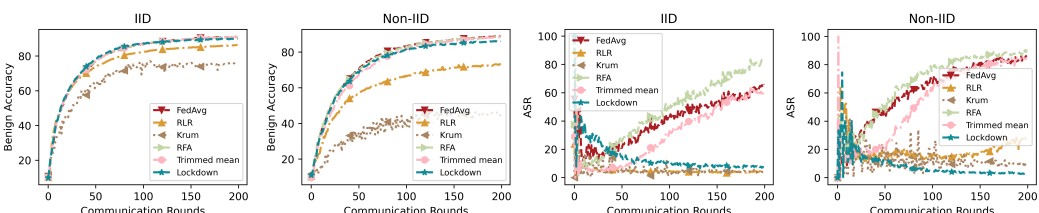

Figure 4: Convergence and defense performance under different defenses.

Table 4: Defense efficacy with varying poison ratio $p$ under CIFAR10.

| Methods | Benign Acc (%) ↑ | | | | | ASR (%) ↓ | | | | | Backdoor Acc (%) ↑ | | | | |
|---|---|---|---|---|---|---|---|---|---|---|---|---|---|---|---|
| (IID) | clean | p=.05 | p=.2 | p=.5 | p=.8 | clean | p=.05 | p=.2 | p=.5 | p=.8 | clean | p=.05 | p=.2 | p=.5 | p=.8 |
| FedAvg | **91.0** | **91.4** | **91.1** | **91.0** | **90.8** | **1.6** | 12.4 | 19.9 | 66.1 | 94.8 | **88.5** | 79.6 | 73.4 | 32.9 | 5.1 |
| RLR | 86.8 | 86.7 | 86.6 | 86.3 | 85.5 | 2.3 | **2.4** | **2.4** | **4.3** | 25.1 | 84.6 | 84.3 | 83.4 | 81.7 | 65.2 |
| Krum | 76.3 | 78.0 | 75.6 | 76.4 | 75.8 | 4.7 | 3.9 | 4.3 | 4.3 | 4.9 | 73.8 | 74.9 | 72.9 | 73.9 | 73.2 |
| RFA | 90.9 | 91.2 | 91.1 | 90.8 | 90.7 | 1.6 | 15.8 | 20.7 | 83.7 | 99.3 | 88.8 | 76.8 | 72.4 | 15.9 | 0.7 |
| Trimmed mean | 91.0 | 90.6 | 91.1 | 90.9 | 90.8 | 1.7 | 5.0 | 20.7 | 61.7 | 96.2 | 88.5 | 84.7 | 72.0 | 36.6 | 3.6 |
| Lockdown | 90.0 | 90.0 | 89.9 | 90.1 | 90.0 | 1.8 | 3.6 | 2.5 | 7.1 | **4.0** | 87.9 | **85.8** | **86.6** | **83.7** | **85.6** |

| Methods | Benign Acc (%) ↑ | | | | | ASR (%) ↓ | | | | | Backdoor Acc (%) ↑ | | | | |
|---|---|---|---|---|---|---|---|---|---|---|---|---|---|---|---|
| (Non-IID) | clean | p=.05 | p=.2 | p=.5 | p=.8 | clean | p=.05 | p=.2 | p=.5 | p=.8 | clean | p=.05 | p=.2 | p=.5 | p=.8 |
| FedAvg | **89.0** | 89.2 | 89.3 | 88.8 | 88.7 | 1.7 | 17.3 | 54.4 | 86.4 | 96.7 | 85.9 | 74.0 | 42.5 | 13.0 | 3.2 |
| RLR | 74.4 | 74.4 | 73.6 | 72.9 | 72.5 | 5.8 | 15.0 | 40.2 | 29.5 | 82.5 | 69.0 | 63.1 | 46.2 | 51.4 | 15.3 |
| Krum | 42.7 | 37.4 | 45.2 | 43.4 | 45.1 | 10.0 | **5.2** | 10.4 | 11.1 | 10.6 | 38.6 | 33.8 | 40.7 | 39.3 | 40.5 |
| RFA | 88.8 | 88.8 | 88.8 | 88.3 | 88.3 | 2.0 | 21.4 | 52.8 | 90.8 | 98.7 | 85.7 | 70.6 | 44.3 | 8.9 | 1.2 |
| Trimmed mean | 88.5 | 88.4 | 88.2 | 88.3 | 88.3 | 1.9 | 25.2 | 48.4 | 84.6 | 96.0 | 85.4 | 67.5 | 47.7 | 14.7 | 3.9 |
| Lockdown | 85.6 | 86.2 | 86.7 | 86.1 | 86.6 | **0.9** | 7.6 | **3.6** | **3.4** | **3.3** | 84.1 | **79.5** | **82.3** | **82.2** | **82.8** |

**Defense efficacy on varying poison ratio.** As shown in Table 4, the Attack Success Ratio (ASR) of Lockdown under different data poison ratios are significant lower compared with vanilla FedAvg without defense (up to 96% reduction), and is also significantly lower than the SOTA backdoor defense solutions. Though ASR of Lockdown is slightly larger than RLR and Krum in IID settings, its benign accuracy is much higher than them (with up-to 5.4% and 15% enhancement comparing RLR and Krum respectively). Another observation is that Lockdown performs better in reducing ASR when the data poison ratio is high (in Non-IID setting, ASR is 7.6% when $p = 0.05$ while ASR is only 3.3 % when $p = 0.8$). This phenomenon is because the subspaces found by the malicious clients will deviate more from benign clients when a larger amount of backdoor samples are injected in their datasets. Therefore, the malicious subspace will overlap less with benign subspaces, resulting in a better isolation, and also benefit the consensus fusion process. In addition, Lockdown significantly advances backdoor accuracy by up-to 79.6% compared to FedAvg without defense, which implies that the Lockdown's model can still recognize the true label of backdoor samples even under attack.

**Defense efficacy on varying attacker ratio.** We fix the poison ratio to 0.5 and vary the attackers ratio to $\{0, 0.1, 0.2, 0.3, 0.4\}$. The results are shown in Figure 5. As shown, Lockdown *consistently* achieves low ASR (at minimum 30% ASR reduction compared to FedAvg when attackers ratio is 0.4), and high benign accuracy in all groups of experiments (at maximum 5% drop of benign accuracy compared to FedAvg). In contrast, RLR and Krum are *sensitive* to attacker ratio and *fail* in defense when attacker ratio is 0.4 (no ASR reduction for Krum, and at maximum 10% reduction for RLR).

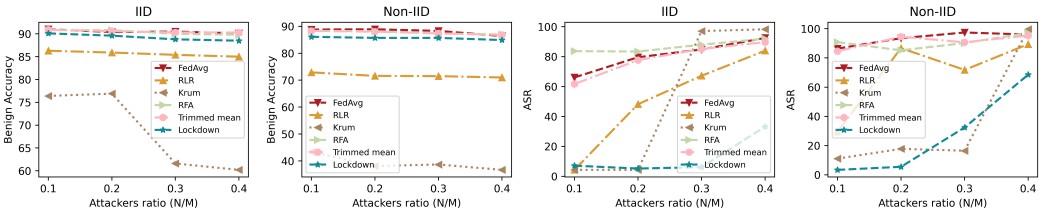

Figure 5: Benign acc/ASR under different attackers ratio.

**Defense against Non-IID extent.** Next we show in Table 5 that Lockdown is robust to different Non-IID extent. Our experiment is done in the default setting, but vary different setting of Dirichlet Parameters $\alpha$, and it shows that Lockdown can defend against backdoor even in extreme Non-IID cases (e.g., $\alpha = 0.2$, which leads to benign accuracy drop of FedAvg from 90.8% to 84.3%). Our another observation is that Lockdown would suffers more benign accuracy loss (respectively 3.7%, 2.7%, 1.2% and 0.7% benign accuracy loss for the four Non-IID levels).

Table 5: Defense efficiency with different Non-IID extent.

| / | $\alpha = 0.2$ | | $\alpha = 0.5$ | | $\alpha = 0.8$ | | IID | |
|---|---|---|---|---|---|---|---|---|
| Methods | Benign Acc ↑ | ASR ↓ | Benign Acc ↑ | ASR ↓ | Benign Acc ↑ | ASR | Benign Acc ↑ | ASR ↓ |
| FedAvg | **84.3** | 74.8 | **88.8** | 86.4 | **89.2** | 74.7 | **90.8** | 66.1 |
| RLR | 64.5 | 82.5 | 72.9 | 29.5 | 82.9 | 24.5 | 85.5 | **4.3** |
| Krum | 26.5 | **2.8** | 43.4 | 11.1 | 43.6 | 6.6 | 75.8 | **4.3** |
| Lockdown | 80.6 | 7.7 | 86.1 | **3.4** | 88 | **3.9** | 90.1 | 7.1 |

**Defense against data-level backdoor.** We simulate attacks with the BadNet, DBA, and sinusoidal method. Our results in Table 6 show that *Lockdown has good generalization ability to data-level*

*backdoor attack.* Overall. Lockdown maintains superior defense efficacy ($< 10\%$ ASR) towards all the three data-level backdoor attacks.

Table 6: Lockdown on data-level attack.

| Methods (IID) | Benign Acc(%) ↑ | ASR(%) ↓ | Backdoor Acc(%) ↑ |
|---|---|---|---|
| BadNet | 90.1 | 7.1 | 83.7 |
| DBA | 90.0 | 2.5 | 86.4 |
| Sinusoidal | 89.5 | 4.6 | 83.6 |
| Methods (Non-IID) | Benign Acc(%) ↑ | ASR (%) ↓ | Backdoor Acc(%) ↑ |
| BadNet | 86.1 | 3.4 | 82.2 |
| DBA | 86.5 | 2.0 | 82.9 |
| Sinusoidal | 87.0 | 2.0 | 81.8 |

Table 7: Lockdown on algorithm/advanced attack.

| Methods (IID) | Benign Acc(%) ↑ | ASR(%) ↓ | Backdoor Acc(%) ↑ |
|---|---|---|---|
| Scaling (Lockdown + NC) | 89.9 | 3.2 | 86.2 |
| Neurotoxin | 89.0 | 2.6 | 86.1 |
| Methods (Non-IID) | Benign Acc(%) ↑ | ASR (%) ↓ | Backdoor Acc(%) ↑ |
| Scaling (Lockdown + NC) | 89.9 | 3.2 | 86.2 |
| Neurotoxin | 86.0 | 1.8 | 83.5 |

**Defense against algorithm-level/advanced backdoor.** Table 7 shows Lockdown's performance on Scaling and Neurotoxin attack. For Scaling attack, the scaling factor is set to 5. We integrate Lockdown with the norm-clipping (NC) in aggregation as proposed in (Sun et al., 2019), and as shown, Lockdown defense is still effective by controlling ASR to <5%. Lockdown is also robust to Neurotoxin, an attack developed from a similar idea as Lockdown (See Appendix A.2 for analysis).

**Defense against adaptive attack.** We design two adaptive attacks that assume the knowledge of Lockdown and try to break it. Both the attacks try to disobey the mask searching process. Specifically, FixMask allows attackers to fix the initial subspace, and Omniscience is able to infer the global subspace that is produced by CF. As shown Table 8, vanilla Lockdown procedure is vulnerable to them. However, we find that there are two methods can be used to rectify Lockdown to accommodate FixMask attack. i) Enlarge the initial pruning/recovery ratio $\alpha_0$, or ii) adopt heterogeneous mask initialization (HM). Both the approaches enhance the dynamics of subspaces, and therefore enhance protection. For Omnisicence attack, the attackers always know consensus subspace and therefore can poison the parameters within it. Lockdown cannot cope with this attack. However, the condition of conducting Omniscience attack is stringent, as it has to acquire either global subspace, or other benign clients' local subspace, meaning that it falls into the *Strong* threat model in Table 1.

Table 8: Lockdown on adaptive attack.

| Methods (IID) | Benign Acc(%) ↑ | ASR(%) ↓ | Backdoor Acc(%) ↑ |
|---|---|---|---|
| FixMask | 89.9 | 63.5 | 35.5 |
| FixMask ($\alpha_0 = 0.1$) | 88.5 | 3.5 | 84.3 |
| FixMask (HM,$\alpha_0 = 0.1$) | 88.5 | 5.8 | 82.8 |
| Omniscience | 89.3 | 44.7 | 52.1 |
| Methods (Non-IID) | Benign Acc(%) ↑ | ASR (%) ↓ | Backdoor Acc(%) ↑ |
| FixMask | 86.8 | 86.3 | 13.1 |
| FixMask ($\alpha_0 = 0.1$) | 85.7 | 4.6 | 79.5 |
| FixMask (HM,$\alpha_0 = 0.1$) | 86.9 | 3.7 | 81.8 |
| Omniscience | 86.8 | 86.3 | 13.1 |

Table 9: Communication and # of parameters for Lockdown under IID CIFAR10 (ResNet9) setting. The communication overhead is the sum of that of $M = 40$ clients in each round.

| Methods | Comm Overhead ↓ | # of params ↓ | Benign Acc(%) ↑ |
|---|---|---|---|
| FedAvg | 2.10GB (1x) | 6.57M (1x) | 91.0 (1x) |
| Lockdown | 0.525GB (0.25x) | 1.643M (0.25x) | 90.0 (0.990x) |

**Communication and model complexity.** As shown in Table 9, we show that Lockdown achieves *smaller communication overhead* (0.25x compared to FedAvg), since only a small subspace of the entire model gradient needs to be sent between server and clients. In addition, the *number of parameters* of the inference model is also *lowered* to 0.25x because the consensus fusion operation prune out the malicious/dummy parameters. Finally, we observe that *benign accuracy only drops by 0.01x* compared to FedAvg, indicating that pruning will not severely perturb the normal function.

**Generalization to varying datasets.** We show our evaluation results on FashionMnist, CIFAR10/100 and TinyImagenet in Table 10. As shown, Lockdown achieves SOTA defense efficacy (compared with FedAvg without defense, with up-to 80.7%, 83.0%, 73.8%, and 93.4% reduction of ASR respectively on the four datasets), and maintains a reasonable loss of benign accuracy compared to FedAvg without defense (with up-to 2.2%, 2.7%, 3.9% and 3.3% drop).

Table 10: Performance on varying datasets.

| / | FashionMnist | | CIFAR10 | | CIFAR100 | | TinyImagenet | |
|---|---|---|---|---|---|---|---|---|
| Methods (IID) | Benign Acc ↑ | ASR ↓ | Benign Acc ↑ | ASR ↓ | Benign Acc ↑ | ASR ↓ | Benign Acc ↑ | ASR ↓ |
| FedAvg | **90.0** | 85.1 | **91.0** | 66.1 | **70.0** | 75.7 | **12.7** | 96.7 |
| RLR | 89.2 | 6.0 | 86.3 | **4.3** | 61.0 | **1.9** | 10.5 | 98.0 |
| Krum | 83.9 | **1.0** | 76.4 | **4.3** | 26.9 | 98.8 | 3.5 | 99.9 |
| Lockdown | 88.7 | 4.4 | 90.1 | 7.1 | 66.9 | **1.9** | 9.4 | **3.3** |
| Methods (Non-IID) | Benign Acc ↑ | ASR ↓ | Benign Acc ↑ | ASR ↓ | Benign Acc ↑ | ASR ↓ | Benign Acc ↑ | ASR ↓ |
| FedAvg | **89.2** | 87.9 | **88.8** | 86.4 | **67.7** | 74.8 | **12.1** | 96.5 |
| RLR | 83.7 | 39.1 | 72.9 | 29.5 | 53.9 | 36.0 | 9.2 | 98.6 |
| Krum | 75.8 | **1.7** | 43.4 | 11.1 | 20.6 | **1.3** | 2.6 | 99.9 |
| Lockdown | 87.1 | 15.6 | 86.1 | **3.4** | 63.8 | 2.5 | 9.6 | **3.1** |

**Visualization**. In Figure 6, We plot each filter's weight of the last convolutional layer of a ResNet-9 model trained with Lockdown. The brighter the color is, the larger the absolute value is, meaning that the filter can be activated by some particular inputs. We find that via our Lockdown procedure, the 86-th filter becomes "suspicious". We plot the weights in the classifier that connects this filter, and accidentally find that this filter contribute most to activate the "horse" neuron, which is our target backdoor label. This illustrates that lockdown can break the "poison-couple effect"– poison parameters (i.e., those in the 86-th filter) only appear in subspace that is not shared by benign clients, which can be effectively filtered out by contrasting the benign and poisoned client's subspace.

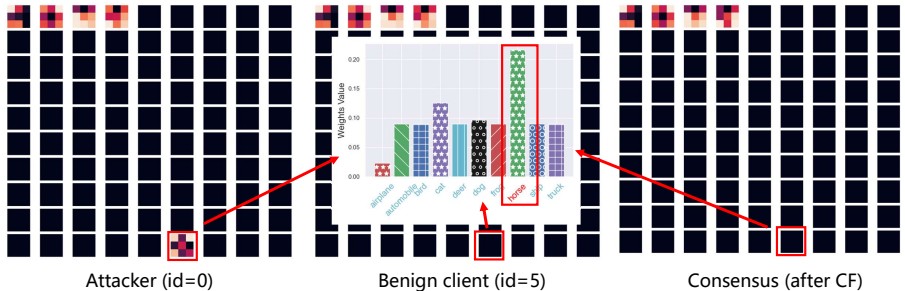

Figure 6: Visualization of Lockdown. Each squre represent weights of a filter. Left/Middle/Right: projecting the weights into an attacker/a benign client/consensus subspace. Bar chart in the middle: weights in the classifier connected 86-th filter with different neurons. "Horse" is our backdoor target.

### 6.3 Ablation and Hyper-parameter Sensitivity Analysis

Ablation study demonstrates the necessity of i) ERK initialization ii) gradient-based recovery, is moved to Appendix A.6. We tune three key hyper-parameters of Lockdown to demonstrate their impacts in our sensitivity analysis. Our findings are: i) setting a proper sparsity $s$ can increase the model's robustness. ii) Setting a proper initial pruning/recovery rate $\alpha_0$ is necessary for effective subspace searching, but too large of it will hurt the model's normal function. iii) Consensus threshold $\theta$ should be set sufficiently large to filter out the malicious parameters. See Appendix A.7 for details.

## 7 Conclusion

In this paper, we study the pruning-based defense for FL and observe a "poison-coupling" phenomenon, which degrades the defense performance. To mitigate such an effect, we propose Lockdown, a backdoor defense based on the idea of isolated subspace training. Empirical evidence shows that Lockdown can significantly reduce the risk of malicious backdoor attacks without sacrificing much on benign accuracy. Future works include studying how to generalize Lockdown to other FL settings, e.g., decentralized FL (Hu et al., 2019; Li et al., 2023a; Shi et al., 2023), personalized FL (Fallah et al., 2020; T Dinh et al., 2020; Huang et al., 2022b, 2023; Dai et al., 2022), etc.

## Acknowledgements

This research is partially sponsored by the NSF CISE grants 2038029, 2302720, 2312758, an IBM faculty award, and a grant from CISCO Edge AI program.

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

# A  Supplementary Material

## A.1  Implementation Details

**Decay pruning rate with cosine annealing.** In our subspace pruning/recovery process, we let the clients prune out $\alpha_t$ percentage of parameters and recover the same amount of parameters to search for the subspace that fits their data. The parameter $\alpha_t$ is decayed with the initial rate $\alpha_0$ with cosine annealing, which can be formalized as follows:

$$\alpha_t = 0.5 \times \alpha_0 \times \left( 1 + \cos\left( \frac{t}{T_{end}}\pi \right) \right) \tag{7}$$

where $t$ is the number of communication round, and $T_{end}$ is the round that the mask searching is ended (notice that $\alpha_{T_{end}} = 0$). In our implementation, we set $T_{end} = T$.

**ERK sparsity initialization.** We use Erdős–Rényi Kernel (ERK) (Evci et al., 2020), an empirical sparsity distribution technique, to distribute sparsity to different layers of a model. Specifically, the active parameters of the convolutional layer initialized by ERK are proportional to $1\frac{n^{l-1}+n^l+w^l+h^l}{n^{l-1}*n^l*w^l*h^l}$, where $n^{l-1}$, $n_l$ $w^l$ and $h^l$ respectively specify the number of input channels, output channels and kernel's width and height in the $l$-th layer. For the linear layer, the number of active parameters scale with $1\frac{n^{l-1}+n^l}{n^{l-1}*n^l}$ where $n^{l-1}$ and $n^l$ are the number of neurons in the $(l-1)$-th and $l$-th layer. ERK initialization, in essence, gives more sparsity to those layers with a larger number of parameters while pruning less on those small layers.

**Subspace pruning.** In the mask searching process, we use parameter's magnitude to guide the pruning of model parameters. We present the PyTorch style code in Algorithm 2 to illustrate the pruning process and, correspondingly, the update of mask. Note that we only prune out parameters that are within the current subspace. Therefore, in line 6, we set the parameters that are out of the subspace to a very large value to prevent from selecting them. After that, we filter out those parameters with the smallest $\alpha_t$ percentage of magnitude and prune them out of the subspace.

---

**Algorithm 2** PyTorch style code for pruning and recovery

---

1: **function  Prune_subspace(** $\alpha_t, \boldsymbol{w}_{i,t,K}, \boldsymbol{m}_{i,t+\frac{1}{2}}$ **)**
2:  Init layer sparsity $\{s_l\}$ given overall sparsity $s$ with ERK
3:  $\boldsymbol{m}_{i,t+1} = \boldsymbol{m}_{i,t+\frac{1}{2}}$
4:  **for** $l = 0, 1, \dots, L-1$ **do**
5:   $num_{prune} = \alpha_t \times$ # of params in the l-th layer
6:   $sort = \text{torch.where}(\boldsymbol{m}_{i,t}^{(l)} == 1, \text{torch.abs}(\boldsymbol{w}_{i,t,K}^{(l)}), 1000\times\text{torch.ones\_like}(\boldsymbol{w}_{i,t,K}^{(l)}))$
7:   $\_, idx = \text{torch.sort}((sort).\text{view}(-1))$
8:   $\boldsymbol{m}_{i,t+1}^{(l)}.\text{view}(-1)[idx[: num_{prune}]] = 0$
9:  **end for**
10:  Return $\boldsymbol{m}_{i,t+1}$
11: **end function**
12:
13: **function  Recover_subspace(** $\alpha_t, \boldsymbol{w}_{i,t,0}, \boldsymbol{m}_{i,t}$ **)**
14:  Derive gradient $\nabla f_i(\boldsymbol{w}_{i,t,0} \odot \boldsymbol{m}_{i,t})$ with one pass of local data
15:  $\boldsymbol{m}_{i,t+\frac{1}{2}} = \boldsymbol{m}_{i,t}$
16:  **for** $l = 0, 1, \dots, L-1$ **do**
17:   $num_{prune} = \alpha_t \times$ # of params in the l-th layer
18:   $sort = \text{torch.where}(\boldsymbol{m}_{i,t+\frac{1}{2}}^{(l)} = 0, \text{torch.abs}(\nabla f_i^{(l)}(\boldsymbol{w}_{i,t,0} \odot \boldsymbol{m}_{i,t})), -1000\times\text{torch.ones\_like}(\boldsymbol{w}_{i,t,K}^{(l)}))$
19:   $\_, idx = \text{torch.sort}((sort).\text{view}(-1), descending=True)$
20:   $\boldsymbol{m}_{i,t+\frac{1}{2}}^{(l)}.\text{view}(-1)[idx[: num_{prune}]] = 1$
21:  **end for**
22:  Return $\boldsymbol{m}_{i,t+\frac{1}{2}}$
23: **end function**

---

**Subspace recovery.** After pruning and before the next round training, we recover the same amount of parameters to explore other parameters outside the subspace. Following (Evci et al., 2020), we use gradient information of the pruned model to guide the recovery process. Here we only recover the

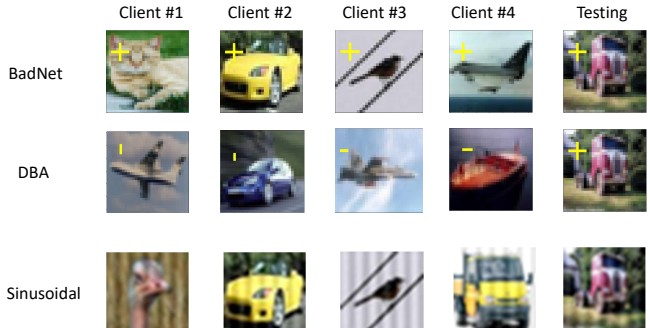

Figure 7: Examples of BadNet, DBA, and Sinusoidal attack. Labels of poison samples are manipulated to the target label (e.g., a horse).

parameters out of the current subspace, and therefore we set the $sort\_value$ of the parameters within the current subspace to a sufficiently small value, as shown in Algorithm 2. Subsequently, we sort in descending to obtain the parameters with the largest-$\alpha_t$ percentage gradient magnitude, and recover them by updating masks.

## A.2 Attack Methods

Table 11: Application of attack methods in threat models. "✓" corresponds to be applicable while "✗" corresponds to be not applicable.

| Attack methods | Threat models | | |
|:---:|:---:|:---:|:---:|
| / | weak | medium | strong |
| BadNet | ✓ | ✓ | ✓ |
| DBA | ✓ | ✓ | ✓ |
| Sinusoidal | ✓ | ✓ | ✓ |
| Scaling | ✗ | ✓ | ✓ |
| FixMask (adaptive) | ✗ | ✓ | ✓ |
| Neurotoxin | ✗ | ✗ | ✓ |
| Omniscience (adaptive) | ✗ | ✗ | ✓ |

As we mention in the main body, we classify the attack model into data-level attack and algorithm-level backdoor. We in the following give brief description of each data-level backdoor that we simulate in federated learning setting.

- **BadNet**. BadNet is the earliest, and also the simplest backdoor attack first proposed in (Gu et al., 2017). To perform BadNet attack, the malicious client simply add the same backdoor trigger on some of the data samples, and modify the label of these poisoned samples to the target label. In test time, the malicious clients can place the backdoor trigger on the test samples, such that the victim model can produce the target output no matter what the original test samples are.

- **DBA.** DBA (Xie et al., 2019) is a backdoor attack specifically targeted on FL. To perform DBA attack, the authors decompose the backdoor trigger into several local pattern, and assign the local pattern to different clients to poison their local data. For test time, the attacker will interpose the completed trigger on top of the test samples they want to manipulate. It is suggested by the authors that DBA is substantially more persistent and stealthy against FL. In our simulation, we decompose the "plus" trigger into 4 local patterns, and let each malicious client to be assigned each local pattern.

- **Sinusoidal attack.** Sinusoidal attack (Barni et al., 2019) shares a similar perspective with BadNet, which also utilize the same trigger for all the malicious clients to poison their samples. However, the backdoor trigger they use is a horizontal sinusoidal signal defined by $v(i, j) = \Delta \sin(2\pi j f / m)$, $1 \leq j \leq m, 1 \leq i \leq l$, for a certain frequency $f$. The authors claim that this design of trigger i) ensures the stealthiness of the attack, but also ii) can be separable with the same (or similar) feature space used by the network to classify the benign samples. In our simulation, we adopt the default hyper-parameter $\delta = 20$ and $f = 6$ for performing this attack.

Examples of these data-level attacks are visually shown in Figure 7.

In the following we give brief description on the algorithm-level backdoor that has been simulated in this paper.

- **Scaling**. The basic idea of Scaling (Bagdasaryan et al., 2020) is to enlarge the gradient update when a malicious client return its update to server. This mechanism allows the malicious client to enlarge its gradient's impact on the global model, and therefore is effective when the poison ratio and attacker number are small.

- **FixMask**. FixMask is an adaptive attack method specifically targeting Lockdown. In Lockdown, the malicious clients are assumed to faithfully search for their subspace using their local data. For FixMask attack, the malicious clients freeze their mask to be the initial mask that is shared by all the clients in round $t = 0$, and refuse to change afterwards.

Particularly, we want to emphasize that the data-level and algorithm-level backdoor can potentially be combined together to produce better attack performance. However, since this paper focus on the defense aspect, we leave a more thorough study of the attack model future work. We also include two advanced attack algorithms that can only be conducted given extra server information in addition to permission on manipulation of the attacker's own training process and data.

- **Neurotoxin**. Neurotoxin proposed in (Zhang et al., 2022b) explores a durable attack method in the scenario that the attackers can only participate limited rounds. Their main observations in the limited participation case are that i) the benign update can recover the global model after attacker ceases attack. ii) the majority of the l2 norm of the aggregated benign update is contained in a small number of coordinates (Let's call these benign coordinates). Utilizing the above observations, the authors propose Neurotoxin, which is to let the malicious clients project their gradient update to the subspace excluding the global coordinates. By this means, the projected updates from the malicious clients are mostly embedded to the coordinates that have less perturbation by the benign updates (which focus on the benign coordinates) after ceasing attack. However, Neurotoxin cannot escape Lockdown defense in principle. There are mainly two reasons. i) Lockdown only broadcast to the clients some coordinates weights (equivalently, some coordinates of gradient update) as per their subspace. Therefore, Neurotoxin cannot obtain the top-k coordinate of the server gradient as benign coordinates. ii) Lockdown requires clients to report the subspace that they want to update, and the subspace that are substantially different from others will be pruned afterwards. In other words, if the attackers adopt neurotoxin to choose the subspace that excludes the benign coordinates, their subspace can be easily identified by comparing with other benign client's subspace, and therefore will be pruned out. In our simulation, we assume Neurotoxin can acquire the server gradient update by some means. Therefore, it is classified as an attack method for *strong* threat model. In our simulation, we set its hyper-parameter mask ratio to be 0.25.

- **Omniscience**. This is an adaptive attack that assumes the knowledge of Lockdown and try to break it. The main idea is to assume the client's has knowledge of the consensus subspace after going through consensus fusion, and project their gradient update into this subspace. This efficiently avoids the malicious weights to be pruned out by the consensus fusion operation. However, the requirement of conducting this attack is very stringent. The malicious client needs to have knowledge of the consensus subspace, which either is leaked from server, or is computed if other clients' subspace is known by the attacker. Neither of this condition is easy to establish for an attacker in a federated learning system.

In summary, we show in Table 11 the attack methods we can perform with specific threat models.

### A.3 Defense Methods

In this section, we give a brief description of the defense baseline we compare against.

- **RLR**. RLR proposed in (Ozdayi et al., 2021) utilizes coordinate-wise server learning rate to inverse the gradient coordinates in which different clients have different sign. Their observation is that the malicious coordinates tends to be those coordinates that have conflicting sign in gradient while for the benign coordinates that are not poisoned, most of the clients will agree with their sign. Therefore, by looking at the gradient update from clients, the server is able to identify the malicious coordinates and subsequently inverse its sign in the aggregation phase. However, the malicious clients are able to launch adaptive attack if he knows the gradient update downloaded from server.

- **RFA**. Aiming at defense against corrupted updates from clients, RFA (Pillutla et al., 2022) utilizes the concept of geometric medium to aggregate the gradient update from clients. Geometric medium avoids the gradient that has excessively large norm (usually is the malicious one) to impact too much on the averaging process. Specifically, when doing aggregation, instead of directly averaging the uploaded gradient, the server aims to obtain global model $v$ that minimizes: $\sum_{i=1}^{m} \|v - w_i\|$, and $w_i$ is the uploaded local model. This problem is solved by the Smoothed Weiszfeld Algorithm. Similar techniques are studied in (Sifaou & Li, 2022), (Ghosh et al., 2019) and (Cao et al., 2020).

- **Krum**. Targeting Byzantine attack, Krum (Blanchard et al., 2017) adopts the idea of finding the gradient update that is closest to its $n - f - 2$ neighbours such that it can ensure $(\alpha, f)$-Byzantine resilience where $\alpha$ is the angle depends on the ratio of the deviation over the gradient , $f$ is the number of attackers. Specifically, Krum aims to find the the $i^*$-th client that minimize $s(i) = \sum_{i \rightarrow j} \|V_i - V_j\|^2$ where $i \rightarrow j$ denotes the set of i'th client's $n - f - 2$ closest neighbours, and $V_i$ denotes the gradient update from client $i$. After identifying $i^*$, Krum returns $V_{i^*}$ as the robust gradient used for aggregation.

- **Trimmed mean**. Trimmend mean is proposed in (Yin et al., 2018) to counter byzantine failures in the distributed machine learning scenario. Their high level idea is to exclude the outlier gradient value when doing aggregation. Specifically, before aggregation, the server coordinate-wise trims the TopK gradient and the bottomK gradient among those uploaded gradient. After trimming, the server assume the outlier has been trimmed, and directly average the clean gradient. In our simulation, we set the trim ratio to be 0.1.

### A.4 Security Analysis

We make the following observations on Lockdown's security performance. Lockdown can successfully defend all the data-level attack, i.e., the attack falls in to the scope of *weak* threat models. For the algorithm-level attacks, we have incorporated an adaptive attack targeting on Lockdown and a gradient scaling method into study. Our results show that Lockdown can also defend all the attacks we have tested. However, since the algorithm-level attacks are more adaptive, we cannot make guarantee that Lockdown is unbreakable by any algorithm-level attacks, especially those that are specially designed for Lockdown. For advanced attack that allows attacker to acquire server's information, we create another adaptive attack Omniscience that can successfully circumvent Lockdown's defense. Performing Omniscience attack needs the attacker to know about the consensus subspace. However, it is challenging, if not impossible, for the attacker to infer the consensus subspace, since only a subset of the server gradient update is distributed to clients, further constraining the global information access of the attackers.

### A.5 More Visualization

**Input-level visualization.** In Figure 8, we add additional experiments to visualize the gradient w.r.t the input of the first layer, which visually explains how different semantic information within the input image contributes to activating the target output neuron.

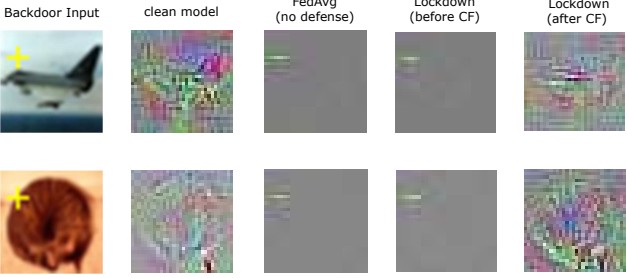

Figure 8: Smooth grad (Smilkov et al., 2017) visualization of models given backdoor input. The first column is data input with backdoor trigger. The subsequent columns demonstrate the gradient with respect to the input of i) a model without being poisoned. ii) a model trained by FedAvg with poisoned data, iii) Lockdown's global model under poisoning before going through consensus fusion (CF) and iv) Lockdown's final model. A clean model emphasize the correct semantic within the input, e.g., wing of a plane, while a poisoned model emphasizes the yellow "plus" backdoor trigger.

**Parameters-level visualization.** In Figure 9, we visualize the projected parameters produced by Lockdown. The experiment is conducted on MNIST with a two-layer MLP model. After reducing its output dimension and reshaping it into the original input, we plot the projected absolute weights of the first layer of MLP. As found, by projecting the global weights into malicious client's subspace (left), the corresponding connectivity that joint the backdoor trigger still present. However, by projecting the global weights into one of the benign client's subspace (middle), the backdoor trigger no longer connects with large absolute weights. The same phenomenon is observed for the consensus subspace after going through consensus fusion (right).

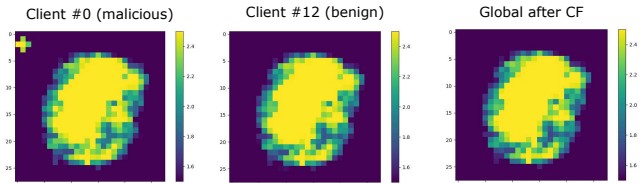

Figure 9: Visualization of absolute global weights after projecting into the local or global subspace. Left: projecting into local subspace of a malicious client. Middle: projecting into local subspace of a benign client. Right: projecting into consensus subspace produced by consensus fusion. The brighter the color is, the feature locates in that part is more important. The bright backdoor trigger "+" is not visible in the middle and right image. See more details in the main text.

## A.6 Ablation Study

We perform ablation study of Lockdown on CIFAR10. BadNet is the default attack method.

**Gradient-based recovery vs. random recovery.** In subspace recovery process, we use gradient magnitude to guide the recovery of parameters. In Table 12, we show the empirical comparison between the gradient-based recovery and random recovery. The results showcase that recovery with the gradient can significantly reduce the ASR (by up-to 78.3% reduction) though the benign acc of the model suffered a little bit (by up-to 2.3% drop). This is because gradient magnitude tends to guide the subspace searching process to acquire heterogeneous subspaces for clients with different training data. With more heterogeneous subspaces, the knowledge transferring between clients will be deterred since their the subspace overlap is small, which leads to the degradation of benign acc. On the other hand, small subspace overlap can also facilitate the process of de-poisoning by consensus fusion, which leads to a reduction of ASR.

Table 12: Ablation study for parameters recovery implementation.

| Methods (IID) | Benign Acc(%) ↑ | ASR(%) ↓ | Backdoor Acc(%) ↑ |
|---|---|---|---|
| Random recovery | **91.1** | 13.0 | 79.7 |
| Recovery w/ gradient (ours) | 90.7 | **1.4** | **87.8** |

| Methods (Non-IID) | Benign Acc(%) ↑ | ASR(%) ↓ | Backdoor Acc(%) ↑ |
|---|---|---|---|
| Random recovery | **88.9** | 17.2 | 74.3 |
| Recovery w/ gradient (ours) | 84.9 | **2.2** | **78.4** |

**ERK initialization vs. uniform initialization.** In the SubspaceInit() function, we use ERK to allocate the sparsity of each layer in a model. To justify the necessity of ERK initialization, we replace the ERK initialization with uniform initialization, which uniformly allocates sparsity to each layer. As shown in Table 13, uniform initialization will largely compromise the benign accuracy and slightly increase the ASR. This justifies that the sparsity should be set larger for the layer with a larger number of parameters (which essentially is what ERK does).

**Consensus fusion (CF).** In Figure 10, we demonstrate the necessity of consensus fusion under different poison ratios. With consensus fusion, benign accuracy is significantly increased by up-to 60% while the ASR is reduced by up-to 80%. This result shows that masking out some malicious/dummy parameters can perturb the backdoor function and thereby curing the poisoned model.

Table 13: Ablation study for sparsity initialization.

| Methods (IID) | Benign Acc ↑ | ASR ↓ | Backdoor Acc ↑ |
|---|---|---|---|
| Uniform | 81.7 | **5.7** | 75.9 |
| ERK (ours) | **90.1** | 7.1 | **83.7** |

| Methods (Non-IID) | Benign Acc ↑ | ASR ↓ | Backdoor Acc ↑ |
|---|---|---|---|
| Uniform | 75.5 | **3.1** | 70.6 |
| ERK (ours) | **86.1** | 3.4 | **82.2** |

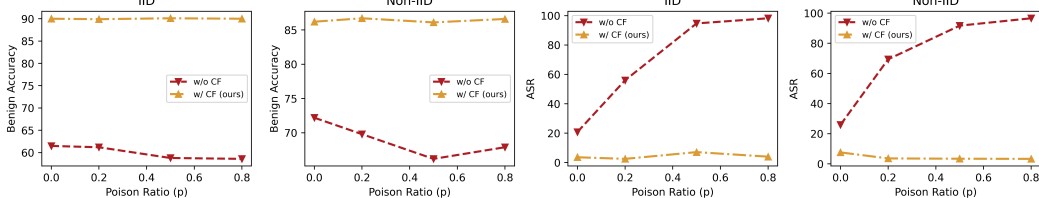

Figure 10: Impact of consensus fusion of Lockdown.

## A.7 Hyper-parameter Sensitivity Analysis

In this section, we perform hyper-parameter sensitivity analysis for lockdown. The evaluation is conducted on CIFAR10 under the default simulation setting in Table 2 unless otherwise specified.

**Sparsity $s$.** In Table 14, we set other hyper-parameters as default and tune the sparsity to different levels. As shown, Lockdown loses its defense efficacy when sparsity is low. This phenomenon is understandable since Lockdown reduces to FedAvg when sparsity is 0. On the other hand, with larger sparsity, the benign accuracy of the model suffers due to the reduction of trainable parameters. Therefore, there exists a tradeoff for the sparsity of Lockdown. Larger sparsity promises lower model complexity, smaller comm overhead, and also lower ASR, but at the cost of losing benign accuracy.

Table 14: Performance of Lockdown under different sparsity $s$.

| $s$ (IID) | Benign Acc ↑ | ASR ↓ | # of params ↓ |
|---|---|---|---|
| 0 | **91.0** | 68.4 | 6.57M |
| 0.2 | 90.9 | 61.1 | 5.26M |
| 0.5 | **91.0** | 10.9 | 3.29M |
| 0.75 | 90.1 | 7.1 | 1.65M |
| 0.9 | 88.3 | **3.0** | **0.66M** |

| $s$ (Non-IID) | Benign Acc ↑ | ASR ↓ | # of params ↓ |
|---|---|---|---|
| 0 | **89.1** | 70.3 | 6.57M |
| 0.2 | 88.4 | 52.6 | 5.26M |
| 0.5 | 87.1 | 14.1 | 3.29M |
| 0.75 | 86.1 | 3.4 | 1.65M |
| 0.9 | 85.0 | **2.9** | **0.66M** |

**Initial pruning/recovery rate.** We also show the effect of initial pruning/recovery for the learning performance. As shown, larger pruning rate would typically results in the drop of benign accuracy but also enhance the ASR under poisoning attack. Specially, when $a_0 = 0$, lockdown reduces to train a sparse subnetowrk from scratch, without evolving the sparse coordinate. This setting cannot eliminate the "poison-couple" effect, therefore the ASR is as high as FedAvg with no defense. On the other hand, setting $\alpha_0$ will also result in isolation of subspace for different clients, resulting in lack of consensus in the global space and therefore leading to drop of benign accuracy.

**Consensus fusion threshold $\theta$.** In Figure 11, we tune the CF threshold $\theta$ to see its impact on different settings of attacker number $N$. In all settings of $N$, we see that: i) $\theta$ should not be set to be too small; otherwise, the benign accuracy would be lower, and the ASR will be higher. ii) $\theta$ also should not be set too large; otherwise, it will severely compromise benign accuracy, but the reduction

Table 15: Performance of Lockdown under different initial pruning/recovery rate $a_0$.

| $a_0$ (IID) | Benign Acc ↑ | ASR ↓ | Backdoor Acc ↑ |
|---|---|---|---|
| 0 | **90.5** | 49.4 | 47.7 |
| 1e-5 | 90.5 | 5.2 | 84.3 |
| 1e-4 | 90.1 | 7.1 | 83.7 |
| 1e-3 | 88.1 | 3.7 | **84.7** |
| 1e-2 | 87.2 | 3.5 | 83.7 |
| 1e-1 | 87.0 | **3.1** | 83.4 |

| $a_0$ (Non-IID) | Benign Acc ↑ | ASR ↓ | Backdoor Acc ↑ |
|---|---|---|---|
| 0 | **88.5** | 85.3 | 14.0 |
| 1e-5 | 87.4 | 8.5 | 78.8 |
| 1e-4 | 86.1 | 3.4 | **82.2** |
| 1e-3 | 84.9 | **2.1** | 80.4 |
| 1e-2 | 83.4 | 5.3 | 76.4 |
| 1e-1 | 83.7 | 5.2 | 77.6 |

of ASR will not be too significant. Per our results, the consensus threshold should be chosen carefully according to the number of attackers, which of course, is unknown in most cases. However, given that the attackers within the system should not take up a large portion, $\theta$ set to be 50% of the total number of clients will be sufficient to counteract the effect of backdoor attack in a general attack scenario.

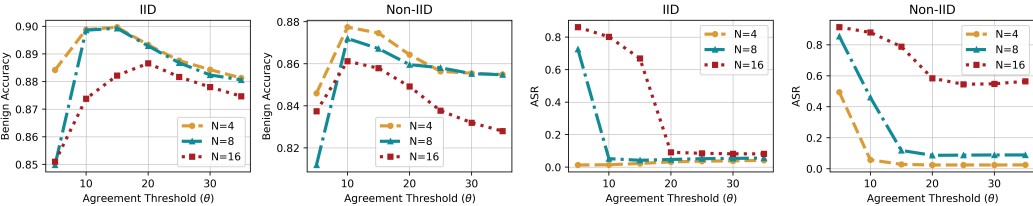

Figure 11: Impact of consensus fusion threshold of Lockdown in different # of attackers setting.

## A.8 Limitations

Our method utilizes sparsity of model to counter backdoor attack. However, we are aware that sparsity in its current stage can hardly guarantee acceleration of the training/inference speed. At present, the current sparse acceleration technique requires 2:4 sparse operation. More specifically, the 2:4 sparse operation requires that there are at most two non-zero values in four contiguous memory, which may not hold for the sparse model produced by Lockdown. But we insist that our method has great potential to achieve truly training acceleration with development of sparse technique.

There are potentially other adaptive backdoor attacks that can break the defense of lockdown, especially under the assumption that attackers have full control over its local training process and has knowledge of the defense. We leave the research of potential attacks against Lockdown as future works.

## A.9 Broader Impact

The poison-coupling effect we discover in this paper might be mis-used to guide the design of backdoor attack method in centralized learning/FL scenario. We will continue this line of research and further propose attack/defense method to better study/mitigate such an effect. We also open-source our code to facilitate researchers/machine learning engineer in academy/industry to study and understand the discovered phenomenon.

