# OpenReview forum: "Lockdown: Backdoor Defense for Federated Learning  with Isolated Subspace Training"
_NeurIPS.cc/2023/Conference — NeurIPS 2023 poster_

### Official Review · Reviewer_t853 · 2023-07-06

**Soundness:** 4 excellent
**Presentation:** 4 excellent
**Contribution:** 4 excellent
**Rating:** 5
**Confidence:** 4

**Summary:**

The paper proposes a pruning-based defense method against backdoor attacks on FL. Specifically, the malicious clients are limited to updating model parameters within an isolated subspace, which reduces the attack surface (or "poison-coupling") for malicious clients as well as the computation/communication cost in FL. Empirical results of a wide range of metrics compared with SOTA baselines have shown the effectiveness of the proposed method.


**Strengths:**

++ It is interesting to detect and defend against backdoor attacks by exploiting model pruning techniques.

++ The observation of "poison-coupling" in FL is inspiring. The paper is well-written, and I enjoy reading the paper.

++ The evaluation involves multiple SOTA baselines of attacking and defense methods.


**Weaknesses:**



W1: Some design assumptions and intuition could be further clarified, see C2

W2: Evaluation could still be improved, see C3

W3: Missing related references

 [1] Zaixi Zhang, Xiaoyu Cao, Jinyuan Jia, Neil Gong. FLDetector: Defending federated learning against model poisoning attacks via detecting malicious clients. SIGKDD 2022

 [2] Hanxi Guo, Hao Wang, Tao Song, Yang Hua, Zhangcheng Lv, Xiulang Jin, Zhengui Xue, Ruhui Ma, and Haibing Guan, Siren: Byzantine-robust Federated Learning via Proactive Alarming, SoCC 2021

 [3] Xiangyu Qi, Tinghao Xie, Ruizhe Pan, Jifeng Zhu, Yong Yang, and Kai Bu. Towards Practical Deployment-Stage Backdoor Attack on Deep Neural Networks. CVPR 2022

W4: minor writing issues

**Questions:**

C1: The paper might violate the double-blind policy---from the Dropbox link provided on Page 4, the files shared show the owner is Huang Tiansheng, https://huangtiansheng.github.io.

C2: Please further clarify the following questions:

* How to defend against malicious clients faking the masks and launching adaptive attacks? Yes, the evaluation has experiments on adaptive attacks, but it would be helpful if the design part discusses the design intuition for adaptive attacks. Attacks like [3] might still be injected into the limited subspace quietly, even with random and heterogeneous mask initialization.

* The intuition for "consensus fusion (CF)"---"Given that those malicious parameters served to recognize backdoor triggers will be deemed unimportant for benign clients, they should not be contained in the subspace of benign clients, which accounts for the majority" should be further discussed and verified. For non-IID cases, such a consensus might not hold.

* Adding more theoretical analysis would be helpful to explain the intuition, though it might not be easy to derive a thorough theory.


C3: Please consider improving the evaluation from the following perspectives:

* More non-IID scenarios could be explored.
* Involving some NLP datasets will be helpful


Writing issues:

* Line 109, "alloww" --> "allows"

* Figure 2 left could be visualized better with the three metrics

* Figure 3 font size is too small

* "For FL, defenses can be classified into two main genres..." The two categories seem not aligned with the classifications in the evaluation (Section 6). For example, which category is RLR belong to?

* "The malicious/dummy parameters should have less chance to appear in benign clients’ subspaces." --- you mean parameters at specific positions?

**Limitations:**

See questions.

---

> ### Author Rebuttal · Authors · 2023-08-03
>
> We thank the reviewer for very detailed comments.
>
> # C1 (Problematic dropbox link)
> We have stopped sharing the problematic Dropbox link. Apology for this un-intensional error.
>
> The original submission used both the anonymous github URL (abstract) and the dropbox URL (page 4). Our intention of sharing this dropbox URL is to open-source the checkpoints such that the difference between centralized and federated backdoor models can be displayed.
>
> The owner id will not be displayed when a dropbox account is logged in. This made us not aware of the fact that the dropbox will expose the owner id. We should have asked other co-authors to double check this dropbox URL.
>
> We sincerely hope that the PC chairs, the reviewers and the area chair will consider that including this dropbpx URL is an unintentional error when making decision. Thank you!
>
>
> # C2+W1  (Design intuition)
> **Q1 (mask faking)** When clients launch adaptive attack by faking the masks, Lockdown remains to be effective, because the core idea of Lockdown is based on the security principle that the malicious parameters (or coordinates) should not be accepted by other benign clients. More specifically, those poisoned coordinates, even though are present in the global model, during FL training, will be pruned out (masked to 0) once Lockdown consensus fusion is performed.  As a result, the poisoned parameters, even injected by adaptive adversary into arbitrary locations of the model, may not evade the consensus fusion operation enforced by Lockdown.
>
> The intuition behind the key ideas of Lockdown:
>
> * **(Intuition of subspace searching)**  Each client will only involve the parameters they deem important in their subspace following the dynamic way of subspace searching.
>
> * **(How consensus fusion works)** By introducing consensus fusion, those parameters that appear less among the subspace will be pruned out. This also indicates that the parameters considered not important by most clients will be pruned out. Given that a majority of the clients is assumed to be benign, the poisoned parameters will not be considered as important ones by the benign clients, and therefore they will be pruned out by consensus fusion.
>
> * **(Role of subspace initialization)** Either heterogeneous or homogeneous subspace (or mask) initialization is empirically observed to work in different degrees in lowering ASR, but we indeed observe that different initialization has impact in the model's benign accuracy.
>
> **(When Lockdown falls)** Our adaptive attack Omniscience can succeed to evade Lockdown by obtaining the full/substantial knowledge of the consensus subspace and inject its malicious parameters within the consensus subspace. However, this attack is very hard to conduct without sufficient knowledge of other clients' subspaces or assuming that the FL server is compromised and collude with the malicious poisoning clients.
>
> **Q2 (consensus fusion in Non-IID setting)**
>
> In non-IID setting, the benign clients would develop more heterogenous subspaces. Hence, when we only reserve the consensus of subspace, the benign accuracy of the model will suffer with a more drastic decline. *However, this does not cause a problem in filtering out the poisoned parameters in conesensus fusion stage*, because there are not backdoor data in the benign clients' local dataset and therefore they would not use the poisoned parameters (or involve them into their subspace), no matter how the non-iid extent is. We show the extra experimental data in our reponse of C3, justifying our claim that Lockdown works well in reducing ASR even in an extreme Non-IID case.
>
> **Q3 (theoretical analysis)**
> One of our future research will be the theoretical analysis of the Lockdown robustness guarantee, including the robustness bound for subspace size and subspace consensus quorum.
>
>
> # C3 + W2 (Additional Evaluation)
> **(More Non-IID experiment)** Following the suggestions by the reviewer under limited time, we did an additional experiment to demonstrate the efficacy of our Lockdown defense on different Dirichlet parameters. Note that $\alpha=0.2$ simulate a highly non-iid extent (the benign accuracy of FedAvg drops a large margin in this case).
>
> The below table shows the number of **benign acc**.
> | $\alpha$    | 0.2  | 0.5  | 0.8  |  iid (infinite) |
> |----------|-------|------|------|-----------------|
> | FedAvg   | 84.3  | 88.8 | 89.2 | 90.8            |
> | RLR      | 64.5  | 72.9 | 82.9 | 85.5            |
> | Krum     | 26.5  | 43.4 | 43.6 | 75.8            |
> | Lockdown | 80.6  | 86.1 | 88   | 90.1            |
>
> The below table shows the number of **ASR**.
>
> | $\alpha$   | 0.2  | 0.5  | 0.8  |  iid (infinite) |
> |----------|-------|------|------|-----------------|
> | FedAvg   | 74.8  | 86.4 | 74.7 | 66.1            |
> | RLR      | 82.5  | 29.5 | 24.5 | 4.3             |
> | Krum     | 2.8   | 11.1 | 6.6  | 4.3             |
> | Lockdown | 7.7   | 3.4  | 3.9  | 7.1             |
>
> As shown, Lockdown reduces ASR to <10\% even when $\alpha=0.2$, which demonstrates its defense efficacy in a highly Non-IID case.
>
> **(NLP dataset)**
> We are working on SST-2 task with a pretrain Bert model.
>
> # W3 (Missing related references)
> We thank the reviewer for sharing the 3 references. We have reviewed them and cited them properly.
>
> # Writing issues
> Thanks for correcting the writing issues. We have corrected the typo as well as the font size issue.  For Figure 2, we now use different marker size to represent the pruning ratio.   Our claim that "defenses can be classified into two main genres" is indeed over-claimed. We now discuss another genre of defenses known as "robust aggregation", which should be the one that Krum and RLR fall in. For the claim "The malicious/dummy parameters should have less chance to appear in benign clients’ subspaces.", here the malicious parameters indeed mean parameters at specific positions, i.e., those positions that are corrupted by the attacker.

---

> > ### Author Response · Authors · 2023-08-10
> > **Discussion welcomed!**
> >
> > Dear reviewer t853,
> >
> > Thanks for your helpful comments, and also for the valuable suggestions on experiements directions!  We have done an experiment on different Non-IID extent. Do you find the results support our claims, and does our explaination coincides with the results? We look forwards to seeing your feedback!

---

> > ### Comment · Reviewer_t853 · 2023-08-19
> >
> > Thanks for the further clarification and experimental results!
> >
> > Some further questions after reading your response and other reviewers' comments:
> >
> > * What is the fundamental cause that leads to poison-coupling effects in FL instead of centralized training?
> >
> > * What are the backdoor attack method, neural network model, and dataset in Fig. 2? How could we justify the "poison-coupling effects" generally exist?
> >
> > * The methodology applied in Fig. 2 to monitor "poison-coupling effects" is still a "black box" approach. Could we treat the FL process as a white box and identify parameters that are important for both benign patterns and backdoor triggers?
> >
> > * I am still not convinced by the intuition "Given that those malicious parameters served to recognize backdoor triggers will be deemed unimportant for benign clients, they should not be contained in the subspace of benign clients, which accounts for the majority"
> >     * If there is coupling, malicious parameters for triggers and benign parameters for normal patterns may share a subset positions in the neural network. Then, malicious parameters could be in the subspace of benign clients.

---

> > > ### Author Response · Authors · 2023-08-19
> > > **Further clarification by authors**
> > >
> > > We thank the reviewer for the insightful questions. Below we try to address them.
> > >
> > > # Clarification of poison-coupling effect
> > > Our initial paper does not articulate well on the poison-coupling effect, which causes a lot of confusion among reviewers. What we actually want to convey is that the poison parameters are statistically coupled with the benign ones in federated learning, making it hard for us to identify and remove them. But we are not trying to claim that the poisoned parameters for triggers and benign parameters for normal patterns share the same subset positions. Existing pruning defense,e.g., CLP  relies on the statistical difference between the benign and poisoned parameters to filter out the poisoned parameters, but this existing method cannot work in a federated learning context, due to the statistical coupling effect we mention.  We will make the point more explicit in the revised version.
> > >
> > > #  Fundamental cause of poison coupling effect
> > > The cause of the poison coupling effect is still not clear in the current stage. We report this phenomenon, hoping that this can raise the awareness among FL community (That's why we want to open-source the checkpoints and the pruning evaluation code in the Dropbox link). The only thing we can tell is that the cause of the statistical difference between poisoned and benign parameters is due to the fact that the poisoned parameters in a poisoned model are usually larger in magnitude than the benign parameters, which explains why the  Lipschitzness constant of poisoned parameters (or channels) are larger.  This is understandable because the poisoned parameters need to enlarge their output when activated by the trigger pattern, in order to overwhelm the output of other benign parameters, which are activated by the benign pattern. However, why the model poisoned by federated learning  does not need to exhibit this statistical feature? We don't have an explicit answer yet unfortunately.
> > >
> > > #  Experimental setting
> > > The experiment is done on CIFAR10 with a ResNet9 model, under BadNet attack, which is the default attack setting we use in the experiment part. We delete this information in our initial submission due to space limitations, but we will add them back in the final version. Thanks for reminding!
> > >
> > > # Justification of the existence of poison-coupling
> > > Because we are trying to say that the benign parameters and the poisoned parameters are coupled in a statistical way. The L2 norm distribution and the Channel Lipschitzness distribution shown in the middle/right of Figure 2 actually justify our claim. We also open-source the checkpoints (though they are temporarily not available due to the Dropbox issue) where everyone can check the federated model, which indeed exhibits poisoned behaviors, but does not show substantial statistical differences between different parameters.
> > >
> > > # Limitation of  "black box" approach
> > > Yes, a more desirable way to convey the message is to get the ground-truth poisoned parameters, and show their statistic of L2 norm/Channel Lipschitzness compared to that of other benign parameters. However, identifying the groud-truth is not easy to achieve due to the astounding size of modern neural networks. We hope the reviewer can understand.
> > >
> > > #  Intuition of the design
> > > To solve the reviewer's concern, we must answer the question: Will malicious parameters for triggers and benign parameters for normal patterns share a subset of positions? Our answer is no. Using our Lockdown method,  the malicious client's final subspace actually has a large overlap with that of the benign clients, and the overlap area actually will be reserved in the consensus subspace yielded by consensus fusion. However, the final model after projecting into the consensus subspace does not exhibit backdoor behavior anymore. That explains that the shared parameters for benign pattern in the consensus are all free from poisoning, and only some unique subset of parameters outside the consensus are dedicated for trigger recognition.
> > >
> > > The connection between our Lockdown design and the poison-coupling effect is that by isolation subspace training, the poisoned parameters are no longer coupled with the benign parameters by looking at the subspaces of all clients (i.e., easier to identify).   We will make this clear in our design motivation.
> > >
> > > We hope this addresses the reviewer's concern, and we are happy to discuss further questions. Thank you for pointing out the points we fail to articulate, which easily leads to confusion.

---

> > > > ### Author Response · Authors · 2023-08-20
> > > > **Does our explanation makes sense to you?**
> > > >
> > > > Hi reviewer t853,
> > > >
> > > > As the discussion ddl is approaching, we just want to make sure that our response really makes sense to you. We are more than happy to discuss if there is still something still unclear. Thank you!

---

### Official Review · Reviewer_ftvs · 2023-07-07

**Soundness:** 3 good
**Presentation:** 3 good
**Contribution:** 3 good
**Rating:** 7
**Confidence:** 4

**Summary:**

The paper proposes a backdoor defense. The method is based on robustly estimating a sparse parameter subspace which is used to restrict updates. Each client will vote for a set of parameters it considers "important" to update, and non-important neurons will be frozen for that iteration. The paper shows that this defense works well empirically and provides an ablation study showing that all of their algorithm's ingredients are necessary.


**Strengths:**

The idea of using sparse training in the context of backdoor defense is interesting. The experiments demonstrate that the method is very effective compared with baseline methods.

Its ablation study also suggests the necessity of individual design choices.

It also evaluates adaptive attacks.


**Weaknesses:**

The paper does not discuss the defense from theoretical aspects. Thus, it is unclear what level of security guarantee can be achieved. Namely, the defense may be only effective because of the current status of research or our knowledge and can be potentially vulnerable to future attacks.


**Questions:**

What security can be achieved by the method?


**Limitations:**

The paper should discuss its limitation on the theoretical side.

---

> ### Author Rebuttal · Authors · 2023-08-03
>
> We thank the reviewer for the very encouraging review and helpful comments. We have provided some security analysis in the supplementary material (see Appendix A.4). A formal verification of the security guarantee is an important research result by itself. It is on our future research agenda. Especially, it is interesting and important to develop the theoretical robustness bound for both the subspace size and the consensus quorum.

---

> > ### Author Response · Authors · 2023-08-10
> > **Discussion welcomed!**
> >
> > Dear reviewer ftvs,
> >
> > Thank you for your strong support of this paper. Just let us know if you need any clarification on the paper after seeing the feedback from other reviewers.

---

> > > ### Comment · Reviewer_ftvs · 2023-08-17
> > >
> > > Thanks for the comments. I have no future questions.

---

### Official Review · Reviewer_dbHt · 2023-07-09

**Soundness:** 3 good
**Presentation:** 4 excellent
**Contribution:** 3 good
**Rating:** 5
**Confidence:** 4

**Summary:**

Federated learning (FL) is a promising approach for privacy-preserving ML applications. However, it is also vulnerable to backdoor attacks. Although some pruning-based methods have been proposed to defend against backdoor attacks, the authors note that it is difficult to prune malicious channels (via Lipschitzness) or weights (via l2 norm) in the federated backdoor setting. The authors named this phenomenon the poison-coupling effect. To address this challenge, the authors propose to isolate the training subspace of individual clients to prevent benign clients from applying malicious clients' parameter updates. The experiment demonstrates that the proposed method can effectively defend against backdoor attacks.

**Strengths:**

1. The poison-coupling effect is an interesting phenomenon and valued problem to be resolved.
2. The idea of defending backdoor in FL with isolated subspace training is interesting and make sense to the problem, and this methods can also reduce the communication cost while defending backdoor attacks.
3. The experiments are comprehensive and seems to be convinced.


**Weaknesses:**

1. This approach relies entirely on the malicious client strictly adhering to the training protocol. That is a strict assumption and hard to be satisfied in the real applications.
2. This approach also relies on some global setting such as count(1{m_{i,t}}) which is the size of the subspace for client i, and the fixed pruning ratio $\alpha_t$ for all clients, which is suitable for all clients with non-iid data set or unbalanced dataset? Those problems is not discussed in this paper.
3. Using gradient sparsification to defend against federal backdoors is a common idea, and the authors do not discuss related work.

**Questions:**

1. My main concern is that Lockdown's performance is entirely dependent on the malicious client strictly following the training protocol. An attacker can easily bypass this defense using two datasets. Specifically, it could first use clean dataset to generate the mask and then performs sparse training on the poisoned training set. I suggest that the defense algorithm should occur mainly on the server, not on the client side.
2. Subspace searching should be the key contribution to solving the poison-coupling effect. However,  the size of the subspace for all clients are same? and the pruning ratio $\alpha_t$ in every round $t$ should be same for all clients? I think the authors should discuss some complicated cases such as Non-IID datasets or unbalanced datasets or multi-task learning etc.
3.  I'd like to start by pointing out that using sparsification methods to defend against attacks and reduce communication overhead is a common paradigm, with similar methods, including Hermes[1] FedMask[2] and SparseFed[3]. However, the authors do not discuss Lockdown's relationship with them. I also note that the proposed method tends to perform worse in terms of benign accuracy. It is not a significant drawback since Lockdown uses only a quarter of the communication overhead compared to FedAvg. Still, I suggest the authors compare the proposed method with other gradient sparsification methods to show that Lockdown can exhibit higher accuracy and robustness for the same amount of communication overhead.
[1] Li, A., Sun, J., Li, P., Pu, Y., Li, H., & Chen, Y. (2021). Hermes: An Efficient Federated Learning Framework for Heterogeneous Mobile Clients. Proceedings of the 27th Annual International Conference on Mobile Computing and Networking, 420–437. https://doi.org/10.1145/3447993.3483278
[2] Li, A., Sun, J., Zeng, X., Zhang, M., Li, H., & Chen, Y. (2021). FedMask: Joint Computation and Communication-Efficient Personalized Federated Learning via Heterogeneous Masking. Proceedings of the 19th ACM Conference on Embedded Networked Sensor Systems, 42–55. https://doi.org/10.1145/3485730.3485929
[3] Panda, A., Mahloujifar, S., Bhagoji, A. N., Chakraborty, S., & Mittal, P. (2022). Sparsefed: Mitigating model poisoning attacks in federated learning with sparsification. International Conference on Artificial Intelligence and Statistics, 7587–7624.


**Limitations:**

see the above

---

> ### Author Rebuttal · Authors · 2023-08-03
>
> We thank this reviewer for the constructive, encouraging, and helpful comments. We below respond to the three weaknesses:
>
> # W1 (Strict assmumption on Lockdown protocal)
> *The answer is NO.* Lockdown defense does not assume that malicious clients have to adhere to the lockdown training protocol. Concretely, we have evaluated the effectiveness of lockdown under two adaptive adversaries (see line 308 page 8): Omniscience and FixMask. Both allow malicious clients to violate the protocol. For FixMask attack, the malicious clients maintain the same initialize masks and refuse to change in the later round(s). Our experiments show that Lockdown remains robust when adopting a proper pruning/recovery ratio. For Omniscience attack, the malicious clients are assumed to have the full knowledge of the consensus subspace, and consequently adapt/adjust their subspace to the consensus. However, to launch a successful Omnoscience attack, the attackers need to obtain (or guess) the accurate consensus subspace, which is hard. It requires to compromise and collude with the FL server. Under such a whitebox Omniscience attack, Lockdown is not robust against poisoning. However, most existing representative poisoning defense methods assume that FL-server is trusted and not colluding with any clients. Due to this reason, we insist that Omniscience attack does not pose severe risk to Lockdown.
>
> #  W2 (Client-level sparsity/pruning ratio)
> Both the size of the subspace for client i and the pruning ratio are the global parameters for all clients under both iid and non-iid setting.
> It is important to note that our lockdown method with these global parameters are easy to implement and yet highly effective, especially under the Non-IID data distribution, which is a representative scenario for federated learning systems with a large population of heterogeneous clients. Hence, in our current Lockdown design, we did not need to consider personalized client-level sparsity of subspace.  One of our ongoing efforts will exam whether the per-client sparsity will offer some value added poisoning defense utility even though it may add more configuration and computation complexity.
>
> # W3 (Relation with previous work)
> We thank this reviewer for 3 related references. Will add them in the related work.
>
> **([1] [2] (sparse training)** both aim to design a communication/computation efficient FL solution. In Hermes, each client employs DNN pruning to learn a personalized sparse DNN and only communicate the updates of the subnetworks with the server. In FedMask, each client will learn a personalized sparse binary mask and share this binary mask with the server while keeping local model unchanged. Instead of learning a shared global model, each client obtains a learned (personalized) binary mask.
>
> Unlike  [1] [2] which optimize per-client DNN pruning solely for communication/computation efficiency, Lockdown designs subspace pruning/recovery/fusion for canceling the poisoning effect at FL server while maintaining the benign local model update quality and the global model accuracy. In this sense, Lockdown differs from the two works in that their design goals are substantially different.
>
> **([3] Gradient sparsification)** utilizes gradient sparsification to defend against data poisoning in FL. Unlike sparse training, gradient sparsification compresses the gradient before sharing with the server. [3] combines global model compression and device-level gradient clipping to defend model poisoning and rely on carefully tuning server side compression ratio and client level clipping. Unlike [3] centering on label flipping attack using gradient compression/clipping, Lockdown is defending against trigger-based backdoor poisoning with the novel and poisoning robust sparse-training techniques.
>
> We thank this reviewer for the helpful comments and useful references.

---

> > ### Author Response · Authors · 2023-08-10
> > **Discussion welcomed!**
> >
> > Dear reviewer dbHt,
> >
> > Are there any issues left by the rebuttal? We are open to discuss if there is still unaddressed concern on the paper. Thanks a lot for improving the overall quality of the paper!

---

> > > ### Author Response · Authors · 2023-08-19
> > > **Can our adaptive attacks eliminate your concern?**
> > >
> > > Hi reviewer dbHt,
> > >
> > > Thanks for the insightful review comment. We believe your main concern lies in our strict assumption of enforcing Lockdown protocol among all the clients, including the malicious ones, which clearly should not be appropriate.
> > >
> > > Do you think our results on adaptive attacks, which assume the attackers to follow other specific strategies in training, help erase your concern? And would you consider updating the overall rating based on our rebuttal? We are also happy to discuss if you have further questions after checking other reviewer's feedback. Appreciate all your efforts putting in reviewing our paper!

---

### Official Review · Reviewer_ozxV · 2023-07-27

**Soundness:** 3 good
**Presentation:** 2 fair
**Contribution:** 3 good
**Rating:** 6
**Confidence:** 3

**Summary:**

The paper addresses the vulnerability of federated learning (FL) to backdoor attacks and the limitations of existing defense solutions in resource-constrained scenarios. It introduces "Lockdown", an isolated subspace training method to counter the poison-coupling effect present in traditional pruning-based defenses for FL. Lockdown modifies the training protocol by isolating subspaces for different clients, uses randomness in initializing isolated subspaces, and employs subspace pruning and recovery to segregate subspaces between malicious and benign clients.
Additionally, quorum consensus is introduced to purge malicious/dummy parameters from the global model.
Empirical results demonstrate that Lockdown outperforms existing methods in defending against backdoor attacks, while also providing communication efficiency and reducing model complexity, making it suitable for resource-constrained FL scenarios.

**Strengths:**

1. Propose a novel defense for FL backdoor attacks. It addresses the limitations of existing defenses in resource-constrained scenarios.

2. Strong and thorough evaluation. The ablation study is also comprehensive.

**Weaknesses:**

I think the paper writing has much room to improve. For example, in Section 4, the paper had better briefly introduce the idea behind channel lipschitzness. Also, in Section 5, although the paper follows a top-down style, it is still full of technical details, without much intuitive explanation or design motivation.

**Questions:**

1. After reading Section 4, I still do not understand what the 'poison-coupling' means. How are the 3 observations in Figure2 connected with (or lead to) the conclusion that 'backdoor parameters tends to be coupled with the benign parameters'? Also what does 'L2 norm of last convolutional layer' imply?

2. Could you elaborate why 'subspace recovery' is necessary? Why not only use 'subspace pruning' to keep important parameters?

3. How does the proposed method improve communication efficiency? What is the key reason? Can the standard model pruning also work?

4. For Table 8, does the communication improvement has something to do with the number of clients?

**Limitations:**

See above.

---

> ### Author Rebuttal · Authors · 2023-08-03
>
> Thanks for the review comments. Below we try to address the reviewer's concern.
>
> # About the poison-coupling effect (Q1)
> **(Poison coupling effect)** We define the poison-coupling effect based on the empirical observation that the parameters used for poisoning by a small percentage of compromised clients in federated learning are relatively harder to identify with high confidence because  the poisoned parameters and the benign parameters are statistically coupled.
>
> **(CLP pruning)** To demonstrate the poison-coupling effect, we measure the effectiveness of the CLP pruning method on both the federated model and the model trained in centralized setting respectively.   CLP's pruning criterion is the estimated Lipscheness constant of each channel. The channel with a larger estimated Lipscheness constant will be pruned because they are more likely to be the poisoned channels.  We have rewritten the CLP pruning in Section 3 to improve the readability.
>
> **(Why 3 observations in Fig. 2 support the poisoning coupling)** The left of Figure 2 shows that if we use CLP to prune the federated model, we will need a larger pruning ratio to safely remove the poisoned parameters and purify the models. This is the first observation, telling that the federated poisoned model is relatively harder to be purified. The middle/right of Figure 2 respectively shows the estimated Lipschitzness constant and the L2 norm of the two poisoned models. We see that the federated model has a more uniform statistical distribution in the channel level in both cases. These two cases (the 2nd/3rd observations) further explain that the poisoned/benign parameters (or channels) are coupled together in a statistical way, and therefore is harder to be identified, i.e., it demonstrates the poison-coupling effect.
>
> **(Implication of the L2 norm of the last convolutional layer)** The L2 norm of the last convolutional layer means the L2 norm of the model parameters within each channel of the last convolutional layer of a model. We only report the statistic data within the last convolutional layer of the model, because the L2 norm exhibits a large discrepancy over different layers.
>
>
> # Necessity of subspace recovery (Q2)
> We introduce the subspace recovery as a necessary subsequent step after subspace pruning because the pure pruning would continuously shrink the federated model round by round, which may lead to numerous errors and/or implementation issues. For example,  when the FL model is pruned during the progress of the training round by round, the pruning ratio will become much harder to configure and tune. This is because the federated model may collapse when the sparsity of the model is larger than a certain threshold. To ensure the robustness of the lockdown defense, we introduce the subspace recovery, which will coordinate the pruning/recovery procedure, making the FL model maintain the same sparsity over the rounds of the FL learning/training process.
>
> # About communication efficiency (Q3)
> The lockdown powered FL training procedure asks every client to train the local model using a subspace structure, and consequently, in each FL round, each of the participating clients only transmits a subset of the full model parameters, i.e., those within each client's own subspace. This can substantially reduce the size of the model parameters shared with the FL server and hence enhance the communication efficiency of federated learning in every round.  In comparison, the standard pruning techniques usually prune the model after it is being trained (like CLP pruning)  when deploying the trained model on edge devices. However, they fail to reduce communication overhead over the iterative federated model training process.
>
> #  Table 8 Communication improvement (Q4)
> In a lockdown powered FL system, the communication improvement is independent of the number of clients. In each round of federated learning, every participating client will gain the communication efficiency under lockdown protection.
>
> In Table  8, we measure the sum of the communication overhead for every client in the FL system. Therefore,  this statistic is related to number of clients.
>
> # Section 5 Intuitive explanation and design motivation (see Weakness section)
> We have updated Section 5 by adding additional illustration on the high-level idea of lockdown solution. The core ideas include:
>
> * **(Subspace searching)**  Each client will only involve the parameters they deem important in their subspace following the dynamic way of subspace searching.
>
> * **(Consensus fusion)** By introducing consensus fusion, those parameters that appear less among the subspace will be pruned out. This also indicates that the parameters considered not important by most clients will be pruned out. Given that a majority of the clients is assumed to be benign, the poisoned parameters will not be considered as important ones by the benign clients, and therefore they will be pruned out by consensus fusion.
>
> We are grateful to the reviewer for the constructive and helpful comments. We sincerely hope that we have addressed your comments and would appreciate the reflection of your satisfaction on the overall rating. Thank you.

---

> > ### Author Response · Authors · 2023-08-10
> > **Discussion welcomed!**
> >
> > Dear reviewer ozxV,
> >
> > We are happy to discuss about the issues left by the rebuttal. Please leave a message if you think something need further clarification.   Thanks for your time!

---

> > > ### Author Response · Authors · 2023-08-18
> > > **Is our writing better to understand with ideas in the rebuttal?**
> > >
> > > Hi reviewer ozxV,
> > >
> > > As you mainly comment on the writing issues, we have try improving by summarizing our high level idea in the rebuttal (which also will be updated in the revised paper). Do you find our explanation of the poison coupling effects, as well as our design motivation makes sense to you, and would you consider updating the overall rating based on our rebuttal? We are also happy to discuss if there is something still causing confusion. Thank you for enhancing the overall quality of the paper!

---

> > > > ### Comment · Reviewer_ozxV · 2023-08-19
> > > >
> > > > Thanks for the explanation. I indeed help me further understand the paper. I suggest the authors to include these clarification into the paper revision.
> > > >
> > > > I still have a question about the subspace pruning/recovery. It seems that the size of $m$ is supposed to be constant throughout the process. Since $m$ is a binary mask, could you explain why the size of $m$ can be keep constant? For example, if some of the parameters to be recovered are not pruned before, isn't the size of $m$ decreasing in this round? Or these non-pruned parameters will be skipped during recovery, and there will be $\alpha_{t-1}$ parameter getting recovered anyway?

---

> > > > > ### Author Response · Authors · 2023-08-19
> > > > > **On weights recovery**
> > > > >
> > > > > # Weights recovery to ensure constant sparsity
> > > > > Yes, the number of non-zero elements of $m$ will be constant in each round. To guarantee that, we indeed cannot recover the parameters that are not pruned before. Therefore, we utilize exactly the same idea, i.e., to skip those non-pruned parameters during recovery, as pointed by the reviewer. This always ensure $\alpha_{t-1}$ parameters getting recovered.
> > > > >
> > > > > To see how we achieve this, you can refer to line 18 of Algorithm 2 in Appendix A.1, where we set the sort_value of the parameters within the current subspace (i.e., those non-pruned parameters) to a sufficiently small value, such that they will never be counted into the recovery number.
> > > > >
> > > > > Thanks again for the review, and we will certainly include the high-level design motivation in the final version!

---

> > > > > > ### Comment · Reviewer_ozxV · 2023-08-19
> > > > > >
> > > > > > Thanks for the authors' explanation. I raise my score to 6.

---

### Decision · Program_Chairs · 2023-09-21

**Decision:**

Accept (poster)

**Comment:**

Most reviewers agree that it is an interesting idea to defend backdoor in FL with isolated subspace training, and detailed empirical evaluation has been provided.  Following the author-reviewer discussion, most concerns raised by reviewers were about requesting additional theoretical analysis and better-clarified design assumptions and intuition. The authors have done an excellent job of addressing most questions. Reviewers generally vote positively after the discussion, though some concerns remain. For example, Reviewer t853 was still unconvinced by the intuition "Given that those malicious parameters served to recognize backdoor triggers will be deemed unimportant for benign clients, they should not be contained in the subspace of benign clients, which accounts for the majority".